# Rapid modulation of choice behavior by ultrasound on the human frontal eye fields

S. Farboud [1] ✉, B. R. Kop [1,2], R. S. Koolschijn [1], S. L. Y. Walstra[1,3], J. P. Marques [1], A. Chetverikov [1,4], W. P. Medendorp [1], L. Verhagen [1,5] & H. E. M. den Ouden [1,5] ✉

A fundamental challenge in neuroscience is establishing causal brain-function relationships with spatial and temporal precision. Transcranial ultrasonic stimulation offers a unique opportunity to modulate deep brain structures non-invasively with high spatial resolution, but temporally precise effects and their neurophysiological foundations have yet to be demonstrated in humans. Here, we develop a temporally precise ultrasound stimulation protocol targeting the frontal eye fields – a well-characterized circuit critical for saccadic eye movements. We demonstrate that ultrasonic stimulation induces robust excitatory behavioral effects. Importantly, individual differences in baseline GABAergic inhibitory tone predict response magnitude. These findings establish ultrasound stimulation as a reliable tool for chronometric circuit interrogation and highlight the importance of neurophysiological state in neuromodulation. This work bridges human and animal research, advancing targeted transcranial ultrasonic stimulation applications in neuroscience and clinical settings.

In recent years, transcranial ultrasonic stimulation (TUS) has emerged as a promising non-invasive technique for brain stimulation, capable of targeting both cortical and subcortical regions with exceptional spatial resolution[1]. This makes TUS highly valuable for studying brain function and offers great potential for therapeutic applications. Much of our current understanding is derived from animal studies[2–6], but there exists a translational gap to human application. The large majority of human studies to date focus on repetitive "offline" protocols with temporally sustained effects[7,8], while those addressing immediate or "online" effects remain limited and often marred by confounds[9] (with the exception of work by Butler and colleagues[10]). This leaves questions on the physiological mechanisms and temporal dynamics of TUS unanswered and calls for robust and replicable protocols in humans. In this study, we introduce an effective online TUS protocol for humans with immediate effects, by leveraging a well-established TUS protocol from non-human primates[2]. To this end, we take advantage of an

evolutionarily conserved brain circuit with a well-characterized link to readily measurable behavior that acts as a model system for more complex decision-making—the frontal eye fields (FEFs).

The role of the FEFs in the planning and generation of saccadic eye movements has been well established in both humans and non-human primates[11,12], and features a basic topographic representation that encodes both the direction and amplitude of saccades in the opposite visual hemifield[11,12]. FEF's involvement in contralateral saccade generation has been further evidenced by lesion studies[13–16] and transcranial magnetic stimulation experiments[17–24]. This well-characterized role of the FEF in contralateral saccades allows for precise characterization of TUS effects. For instance, in macaques online TUS of the FEF was found to bias saccades towards the contralateral side, which suggests that stimulation has a net excitatory effect[2]. However, it is not known whether these results can be directly translated to humans, i.e., whether online TUS stimulation of the human FEF can induce the same

[1]Donders Institute for Brain, Cognition and Behaviour, Radboud University, Nijmegen, The Netherlands. [2]School of Medicine, Stanford University, Stanford, CA, USA. [3]Integrative Model-Based Neuroscience Research Unit, University of Amsterdam, Amsterdam, The Netherlands. [4]Department of Psychosocial Science, Faculty of Psychology, University of Bergen, Bergen, Norway. [5]These authors contributed equally: L. Verhagen, H. E. M. den Ouden. ✉e-mail: soha.farboud@donders.ru.nl; hanneke.denouden@donders.ru.nl

excitatory effect, or whether anatomical, physiological, and behavioral differences between humans and non-human primates would instead result in net inhibitory or perturbatory effects. Indeed, the effect of TUS has been found to vary from excitatory to inhibitory to perturbatory depending on the specific stimulation protocol settings[25], underscoring the need for caution when interpreting TUS-induced behavioral changes.

Considering that the effects of brain stimulation are highly dependent on brain states and traits[26–30], it is expected that TUS effects vary not only between species, but also between individuals. Consequently, it is pertinent to consider the individual neurophysiological state when investigating the mechanisms and consequences of TUS. This interindividual variability may be influenced by factors such as an individual's cortical inhibitory tone, which has been shown to impact the effects of other noninvasive stimulation methods[31]. Moreover, differences in cortical inhibitory tone in the FEFs have been linked to individual variations in the capacity to resist distractions while generating saccades[32]. Therefore, it seems likely that the neuromodulatory effects of TUS on an individual may have different effects. Given that TUS may modulate both excitatory and inhibitory neuronal populations in the brain, we hypothesize that the net effects of TUS could be shaped by individual differences in the excitation/inhibition balance. To explore whether interindividual differences in the effects of TUS are similarly inhibitory tone-dependent, we measured individual level concentrations of the inhibitory neurotransmitter GABA+ in the FEF using magnetic resonance spectroscopy (MRS).

In the present study, we tested the hypothesis that TUS applied to the human FEF has an immediate, excitatory effect on saccade direction, and that this effect is mediated by local inhibitory tone. Participants completed a simple saccade choice task while receiving TUS during stimulus presentation, applied to either their left or right FEF ("stimulation"), or to the left or right hand motor cortex (M1) ("active control"). FEF TUS induced a significant increase in the selection of contralateral saccades, directly replicating findings from a previous study in macaques[2] and indicating that FEF TUS has net excitatory effects on saccade selection in humans. Notably, participants' characteristic inhibitory tone in FEF was found to predict inter-individual differences in the effect of TUS, suggesting that TUS susceptibility is linked to an individual's inhibitory tone. Taken together, our findings pave the way to use TUS as an effective and temporally specific tool to study the functional circuit dynamics of the human brain and offer critical insights into the factors that drive interindividual differences in response to this neuromodulation technique.

## Results

### Baseline saccade task behavior

Thirty-five right-handed participants ($M_{age}$ = 24.1, $SD_{age}$ = 3.2, range = 20–32; 15 females, 20 males) performed a saccadic decision task in which two visual stimuli were presented asynchronously and equidistantly on either side of fixation (Fig. 1A). Participants were instructed to saccade as quickly as possible to the stimulus that appeared first (i.e., target). We examined the probability of making a rightward saccade across all stimulus onset asynchronies (SOA, i.e., delay between target and distractor). Participants performed well on the task in the baseline (sham) condition: When the target is on the right, participants were more likely to make a rightward saccade ($b$ = 14.1, 95%-CI [13.0, 15.4], $\chi^2(1)$ = 530, $p$ < 0.001, Fig. 1B). At the group-level, there is a lack of a noticeable rightward or leftward bias in the sham condition, although within participants there is variability in baseline side bias (Fig. 1B).

In contrast to magnetic or electrical stimulation, which can directly elicit evoked responses (motor-evoked potentials[33]; visual-evoked potentials[34,35]), TUS at the low intensities employed in the current study is thought to act through modulating sub-firing threshold, ongoing activity, thereby gently nudging network dynamics rather

than producing immediate, large-scale neural responses[36]. Therefore, we expected TUS effects to bias responses primarily on trials with short SOAs, i.e., high uncertainty about which target appeared first. On these trials, with low sensory evidence for left- or rightward responses, a small perturbation has the potential to bias the outcome, and thus we expected that TUS could tip the scales[2]. In contrast, for long SOAs there was low uncertainty about the correct response and TUS would not be able to overrule the associated robust neural signal. Therefore, we designed the task to oversample trials with shorter SOAs (Fig. 1C) and focused the primary analysis on trials with SOAs where participants were <75% correct (Fig. 1B, hereafter referred to as the choice domain).

### FEF-TUS shows contralateral bias correlated with GABA+ levels

Ultrasonic stimulation of both the left and right FEFs significantly increased contralateral saccades (Fig. 2C,D; $b$ = −0.25, 95%-CI [−0.40, −0.10], $\chi^2(1)$ = 10.3, $p$ = 0.001). Left vs right FEF increased the odds of a rightward saccade by 13% ($OR$ = 1.13, 95%-CI 1.05–1.22), corresponding to Cohen's d (logit) = 0.07 (95%-CI 0.03–0.11). At the participant level, the within-subject SMD of model-predicted probabilities was dz = 1.27 (Hedges' $g$ = 1.24; $n$ = 35), indicating a consistent modulation across subjects. This finding aligns with our hypothesis that the protocol induces excitatory behavioral effects, and replicates prior findings observed in non-human primates[2]. This excitatory behavioral effect on contralateral saccades was not observed for stimulation to left versus right M1 (details reported below; for visual comparison of FEF-sham and FEF-M1 refer to Fig. S1). Comparison of each FEF condition relative to baseline (sham) showed that the sham condition provided a baseline between left and right FEF TUS effects, whereby left FEF TUS increased the proportion of rightward saccades ($b$ = −0.12, $p$ = 0.023), and right FEF showed a trend toward more leftward saccades ($b$ = 0.12, $p$ = 0.08). These results highlight the specificity of the effects to the FEFs and provide converging evidence of direct TUS-induced behavioral changes in humans.

There was substantial interindividual variability both in baseline (sham) directional bias (Fig. 1B) as well as in the susceptibility of saccade direction to TUS stimulation (Fig. 2C). Therefore, we next asked whether the baseline neural inhibitory tone in participants' FEF could explain interindividual differences in TUS susceptibility. Note that we measured only left hemispheric MRS (in FEF and M1, for details see "methods"). We found that changes in saccade bias induced by left FEF TUS relative to sham were predicted by baseline FEF GABA+ levels (condition (left FEF/sham) x FEF GABA+: $b$ = −0.21, 95%-CI [−0.39, −0.04], $\chi^2(1)$ = 5.6, $p$ = 0.017; Fig. 3A). Specifically, higher baseline GABA + levels in the left FEF were associated with a stronger rightward bias on sham trials (sham x FEF GABA+: $b$ = 0.14, 95%-CI [0.04, 0.24], $\chi^2(1)$ = 7.0, $R^2$ = 0.096, $p$ = 0.008; Fig. 3B, top). Importantly, following TUS stimulation, this relationship of baseline GABA+ and rightward bias disappeared (left FEF x FEF GABA+: $b$ = −0.08, 95%-CI [−0.22, 0.07], $\chi^2(1)$ = 1.1, $p$ = 0.3; Fig. 3B, bottom). Thus, TUS increased contralateral responding predominantly in participants with lower baseline GABA+ levels in the FEF (voxel placement: Fig. 3C).

### M1 TUS does not affect saccade choices

To ensure that the observed effects of TUS on saccade direction were specific to the FEF, in half of the stimulation trials the left and right M1 (hand area) were stimulated as control regions. Similar to the FEF stimulation, we used an fMRI functional localizer to determine the participant specific target for M1 TUS (Fig. 4A). Again, post hoc acoustic wave propagation simulations confirmed that we successfully targeted these regions (Fig. 4B). As expected, left and right M1 TUS did not induce significant differences in contralateral saccades, further supporting the specificity of the observed effects to the FEF and ruling out potential confounds ($b$ = −0.09, 95%-CI [−0.22, 0.04], $\chi^2(1)$ = 1.8, $p$ = 0.12; Fig. 4C).

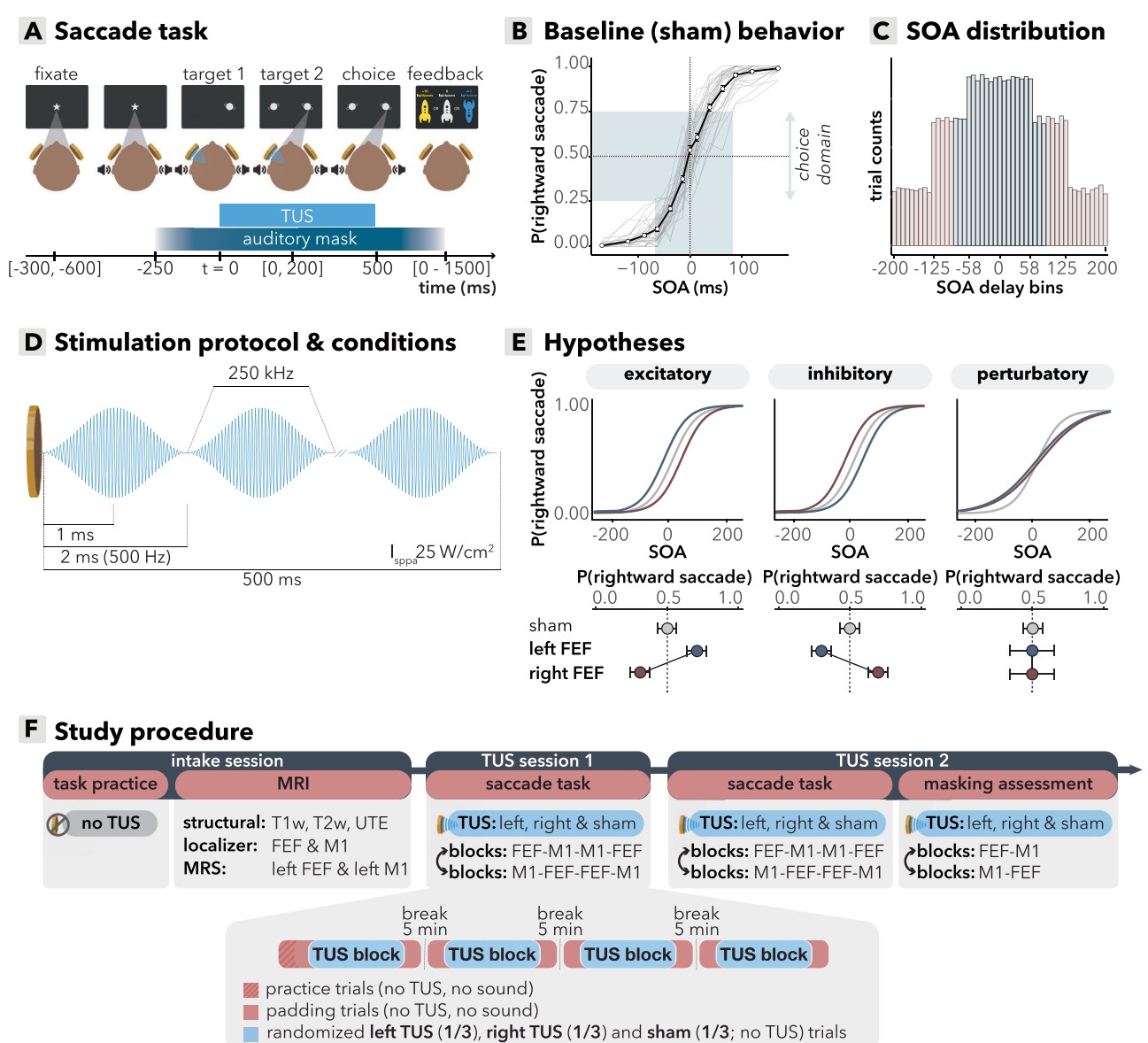

**Fig. 1 | Study design and baseline behavioral results. A** Task design. Two later-alized targets appeared, spaced by a variable SOA (0-200 ms). Participants (*n* = 35) made saccades to the first target and received feedback (correct: +10 points; incorrect: −10; correct but late: 0). TUS was applied to left or right hemisphere, paired with a masking sound; sham trials consisted of auditory mask only. **B** Baseline behavioral performance followed a typical psychometric sigmoid function, with lower accuracy at short SOAs. Because TUS effects were expected to be most apparent under low sensory evidence, we defined the choice domain as delays at which performance was <75% correct (blue shading). Black line indicates group mean; error bars indicate standard error of the mean across participants; gray lines represent individual participants. Inferential statistics used continuous SOAs, but for visualization data is binned into approximately equal-trial bins with SOA intervals of [0], [8.3−25.0], [33.3−50.0], [58.3−75.0], [83.3−100.0], [108.3−141.7], [150−200]. **C** SOAs frequency, with shorter delays oversampled to increase sensitivity to biasing effects under high uncertainty. Blue bars indicate the empirically defined average choice domain. **D** TUS was delivered for 500 ms per trial, starting at onset of the first target (fundamental frequency 250 kHz; pulse repetition frequency 500 Hz; intensity in free water $I_{SPPA}$ = 25 /cm²), with 1-ms ramp-up and ramp-down per pulse. Conditions comprised left/right FEF, left/right M1, and sham. **E** Hypothesized TUS effects. Predicted behavioral patterns if TUS induces a (i) net excitatory bias (more contralateral saccades), (ii) net inhibitory bias (more ipsilateral saccades), or (iii) perturbatory effect (reduced accuracy/increased noise, but no bias). Note that this panel visualizes predicted patterns and does not represent empirical data. **F** Participants completed one intake and two TUS sessions. Intake included task practice, structural MRI for neuronavigation, functional localizers for individual targeting, and baseline GABA + MRS (left FEF and M1 only). Each TUS session comprised four blocks of the saccade task with coun-terbalanced FEF/M1 block order. A masking assessment at the end of the final session evaluated whether participants could detect stimulation presence/side under auditory masking. Source data are provided as a Source Data file.

However, at delay 0, where both targets appear on the screen simultaneously, we observed a qualitative difference in rightward saccade probability in M1 stimulation (Fig. 4D) that mirrors the pattern of FEF stimulation. We remark that these zero-delay trials are qualitatively different from the delay trials because of simultaneous target presentation, which may be easily identified and thus trigger different cognitive processes[37,38]. This observation is further explored in the

Supplementary Information (Supplementary Information S.1). For consistency across all analyses, we selected the choice domain data without these delay-0 trials.

Critically, a formal side-by-region (FEF/M1) comparison revealed a significant interaction (*b* = 0.23, *95%-CI* [0.03, 0.43], $\chi^2(1)$ = 5.1, *p* = 0.025) between stimulation side (left/right) and region (FEF/M1). The side-by-region model demonstrated stronger side modulation in

### Individual FEF localization and targeting

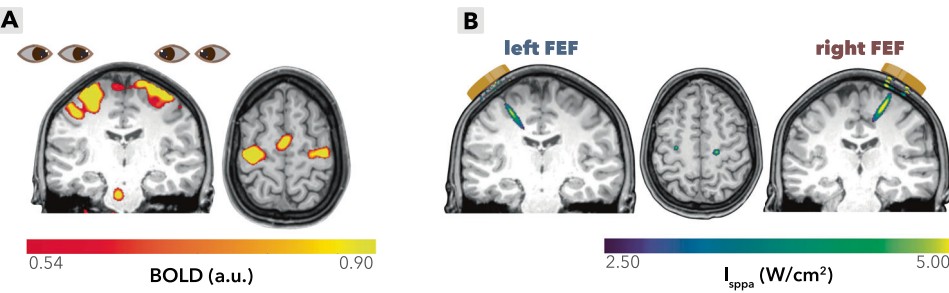

### TUS of the FEFs results in increased contralateral responses

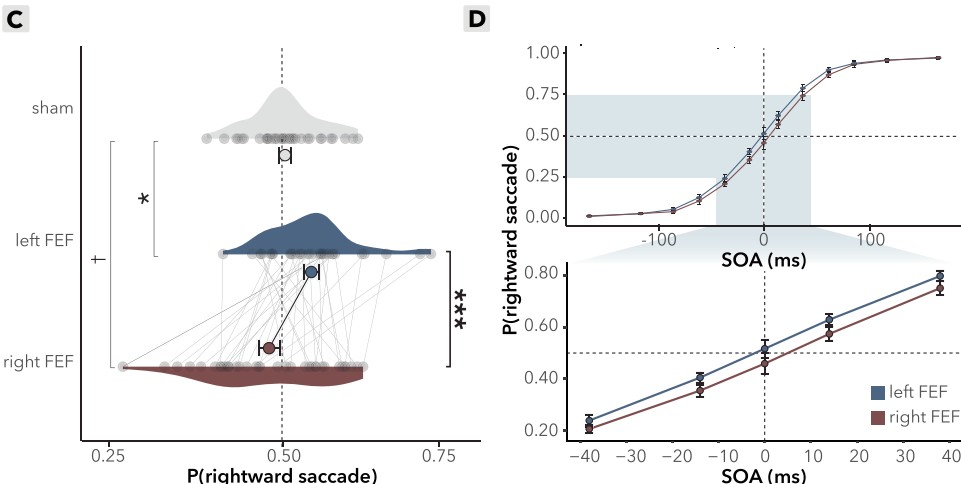

**Fig. 2 | Main behavial TUS effect. A** Individual FEF localization. BOLD responses of left and right FEF for a single subject. Participants performed a functional localizer during the intake session in which they alternated between left/right saccades and fixation, allowing for individual localization of the FEFs. **B** Acoustic simulation of TUS wave propagation. Acoustic simulations of left and right FEF for a single subject are shown. The simulation depicts the estimated intracranial intensity ($I_{sppa}$) with an intensity cutoff at the full width at half maximum (FWHM). **C** FEF TUS effects ($n = 35$). Choice-domain average effects. Gray dots represent individual participants' mean saccadic directions within the choice domain. Colored dots represent group means, error bars indicate the standard error of the mean (S.E.M.). ***$p < 0.001$, *$p < 0.05$, †$p < 0.1$. **D** FEF TUS effects ($n = 35$). Stimulation of the left and right frontal eye fields (FEF) led to increased contralateral saccades, particularly

within the choice domain (highlighted in light blue, bottom). Compared to each other, left FEF stimulation produced more rightward saccades, while right FEF stimulation led to more leftward saccades. Data are binned for visualization purposes into bins of approximately equal numbers of trials, resulting in SOA intervals of [0], [8.3–25.0], [33.3–50.0], [58.3–75.0], [83.3–100.0], [108.3–141.7], and [150–200]. Bins are symmetric for negative values. Note that all inferential statistics were performed using continuous SOAs. Dots represent the group mean per bin, and error bars indicate the S.E.M. across participants. **C**, **D** Source data are provided as a Source Data file. Exact $p$ values are presented in the main text. Statistical significance was determined using two-sided logistic mixed effects regressions. No multiple comparisons were applied.

FEF than in M1 by 26% (ROR = 1.26, 95%-CI 1.03–1.56), corresponding to Cohen's d (logit) = 0.13 (95%-CI 0.01–0.24) indicating a small but reliable interaction. This indicates that the TUS effects influencing choice bias and saccade behavior are specific to FEF stimulation and not to M1 stimulation, and excludes the possibility that these effects are driven by confounds such as auditory or somatosensory stimulation. This finding reinforces the conclusion that TUS selectively modulates behavior via its impact on the FEFs. Finally, between-variability in the effects of TUS in M1 in saccade direction could not be explained by subjects' baseline GABA+ levels in M1 (condition (left M1/sham) x M1 GABA+ : $b = 0.06$, 95%-CI [−0.14, 0.26], $\chi^2(1) = 0.3$, $R^2 = 0.1$, $p = 0.5$; Fig. 5A–C). These findings underscore the specificity of the TUS effects to the FEFs and provide additional evidence against potential confounds in the study design.

### Control and follow-up analyses

**TUS does not disrupt overall performance.** In order to assess potential perturbatory effects of FEF TUS on performance, we completed a regression with "correct response" on TUS and sham trials in the FEF blocks as dependent variable and side, region and delay as independent variables. Note that again zero-delay trials are excluded

from this analysis because no correct response can be defined. For a binary choice task, sensory noise is directly reflected in the overall accuracy (i.e., the slope of the psychometric curve is inversely related to the variance of the underlying signal probability distribution). There was no perturbatory effect of TUS on performance (TUS vs. sham: $b = 0.07$, 95%-CI [−0.13, 0.27], $\chi^2(1) = 0.5$, $p = 0.5$).

Additionally, we performed supplementary analyses (Supplementary Information S.2) examining estimation of bias, including effect size in decision curve shift (horizontal bias), slope, and lapse rate to confirm that the observed TUS effects were specific to bias and not confounded by changes in slope or lapse rate.

**Online TUS effects are immediate and short-lived.** Having demonstrated that TUS of FEF has an excitatory effect and that this effect is specific to stimulation of FEF, we next assessed the duration and persistence of TUS effects on saccade direction. This is critical to characterize the temporal dynamics of ultrasonic neuromodulation, in terms of how fast effects arise, and whether they persist into the next trial. Slow and sustained effects suggest early-phase plasticity mechanisms to drive the observed behavior, while fast and temporally precise effects suggest modulation of spiking activity. The latter would

**Baseline FEF GABA levels predict interindividual variability in FEF TUS effects**

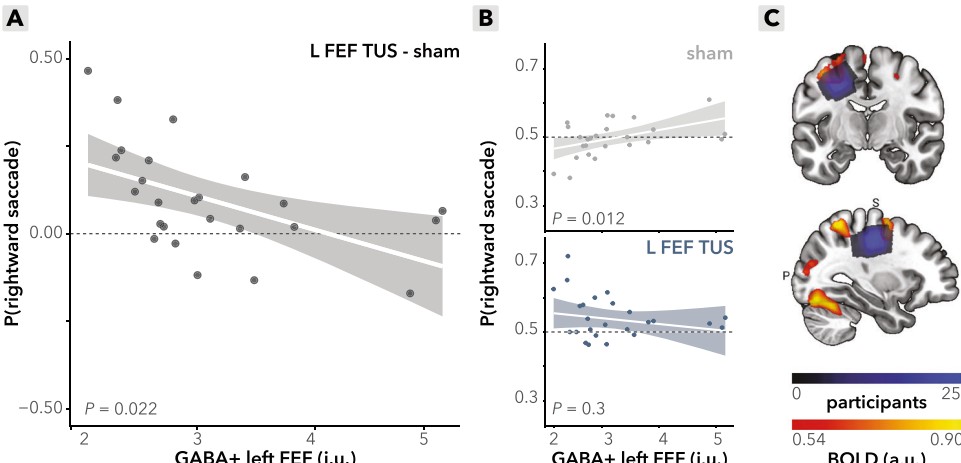

Fig. 3 | **Interindividual variability in FEF effects. A** FEF GABA+ predicts FEF TUS effects. The relationship between baseline GABA+ levels in the left FEF and the effect of TUS on saccadic bias, calculated as the difference in probability of making a rightward saccade between left FEF TUS and sham conditions. Higher baseline FEF GABA+ levels correlate with a weaker TUS effect on rightward saccades ($p = 0.022$). The line is a linear fitted group mean with a 95% confidence interval around the fit; each dot represents a participant. **B** FEF GABA+ predicts baseline saccade behavior. Top: baseline left FEF GABA+ levels significantly correlate with rightward saccade probability during sham stimulation alone ($p = 0.012$). Bottom: under left FEF TUS, this correlation with rightward saccade probability is absent ($p = 0.3$). The line is a linear fitted group mean with a 95% confidence interval around the fit; each dot represents a participant; each dot represents a participant. **C** MRS voxel placement. Magnetic resonance spectroscopy (MRS) voxel placement for measuring GABA+ concentrations in the left frontal eye field (FEF). Color overlays represent GABA+ concentration distributions in each region. **A**, **B** Source data are provided as a Source Data file. Exact $p$ values are presented in the main text. Statistical significance was determined using two-sided binomial linear mixed effects regressions. No multiple comparisons were applied.

enable TUS to be used for cognitive chronometry: to disentangle the functional contributions of brain regions and circuits across time.

First, we examined whether there were any carryover effects of stimulation on sham trials that followed TUS trials. First, when analyzing sham trials during FEF blocks, no significant carry-over effects were observed. More specifically, saccade direction was not affected by the side stimulated on the preceding trial ($FEF_{t-1}$ (left/right): $b = 0.15$, 95%-CI [−0.07, 0.37], $\chi^2(1) = 1.8$, $p = 0.18$; $BF_{01} = 2$). However, when pooling together sham trials during FEF and M1 blocks, participants made significantly more ipsilateral saccades on these trials, directing their saccades toward the side stimulated on the preceding trial ($side_{t-1}$: $b = 0.10$, 95%-CI [−0.18, −0.02], $\chi^2(1) = 5.3$, $p = 0.021$). Crucially, this ipsilateral bias did not differ between FEF and M1 stimulation ($side_{t-1}$ x region $_{t-1}$: $b = 0.02$, 95%-CI [−0.06, 0.10], $\chi^2(1) = 0.3$, $p = 0.6$, $BF_{01} = 12$; Fig. 6A), and can therefore not explain the observed specific effects on contralateral saccades following FEF (but not M1) TUS. In Supplementary Information S.3, we will briefly further discuss the non-specific (potentially attentionally driven) after-effects.

Finally, to assess the immediacy of TUS effects relative to stimulation onset, we quantified TUS effects on the fastest saccades, defined as trials with a saccade reaction time below the median of 265 ms. Even on this subset of trials where participants received less than 265 ms of stimulation prior to saccade onset, TUS significantly shifted saccade direction contralaterally (FEF (left/right): $b = −0.32$, 95%-CI [−0.59, −0.07], $\chi^2(1) = 6.1$, $p = 0.013$). In contrast, no significant saccade bias was observed for left versus right M1 stimulation (M1 (left/right): $b = −0.15$, 95%-CI [−0.38, 0.07], $\chi^2(1) = 1.8$, $p = 0.19$). Taken together, our results highlight the specificity and speed of TUS effects on saccade direction, reinforcing that they are immediate, fast and specific to the FEF.

**Masking assessment.** To estimate the potential impact of auditory or somatosensory confounds[9,39–42], we included a masking assessment at the end of the second TUS session (Fig. 1F). This assessment allowed us to verify that potential confounds could not explain the observed dissociation of TUS effects over FEF versus M1. Participants received

stimulation (or sham) repeatedly either over FEF or M1 (in blocks), all with an auditory mask, and reported (i) whether they perceived stimulation, and (ii) on what side (forced choice, left vs. right). First, participants reported perceiving stimulation more frequently on TUS trials compared to sham (stimulation (TUS/sham): $b = 2.7$, 95%-CI [1.9, 3.4], $\chi^2(1) = 53.6$, $p < 0.001$). Crucially, however, this ability to detect TUS versus sham did not differ between conditions (region (M1/FEF): $b = 0.25$, 95%-CI [−0.36, 0.86], $\chi^2(1) = 0.7$, $p = 0.4$, $BF_{01} = 4.1$; Fig. 6B). Second, on TUS trials, participants were biased to report perceiving stimulation *contralaterally* to the side of actual stimulation (side: $b = −1.2$, 95%-CI [−2.1, −0.3], $\chi^2(1) = 7.$, $p = 0.008$, $BF_{01} < 0.001$; Fig. 6C). However, again this contralateral reporting bias was not significantly different between FEF and M1 stimulation (side (left/right) x region (FEF/M1): $b = −0.5$, 95%-CI [−1.5, 0.5], $\chi^2(1) = 1.1$, $p = 0.3$). In Supplementary Information S.3, we further discuss of this contralateral TUS perception in the context of lateralized non-specific aftereffects reported above. Taken together, while putative confounding factors were present in our study, their effects were present in both FEF and M1 conditions, and thus crucially cannot account for our main findings. More broadly, this masking assessment confirms the presence of putative confounding factors and emphasize the importance of active control conditions for online TUS protocols.

## Discussion

This study provides evidence for effective online transcranial ultrasound stimulation (TUS) on saccadic decision-making in humans. These results advance our understanding of the underlying neural mechanisms contributing to interindividual differences. We found that short TUS pulse trains (500 ms) to the FEF but not the primary hand motor cortex (M1) have immediate, short lived effects promoting contralateral saccades. Importantly, the effect of FEF TUS is associated with individual inhibitory tone, as indexed with MRS. These findings provide behavioral and neurophysiological evidence for the direct effects of TUS on human brain function, establishing its potential as temporally specific neuromodulatory tool for advancing fundamental neuroscience and enhancing our understanding of temporally dynamic brain-behavior relationships.

### Individual M1 localization and targeting (control region)

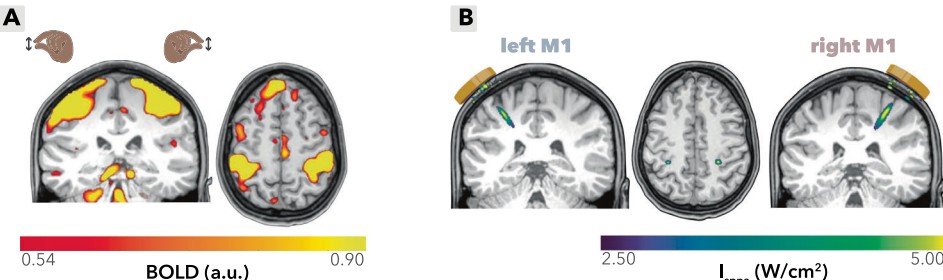

### M1 TUS does not influence saccade direction

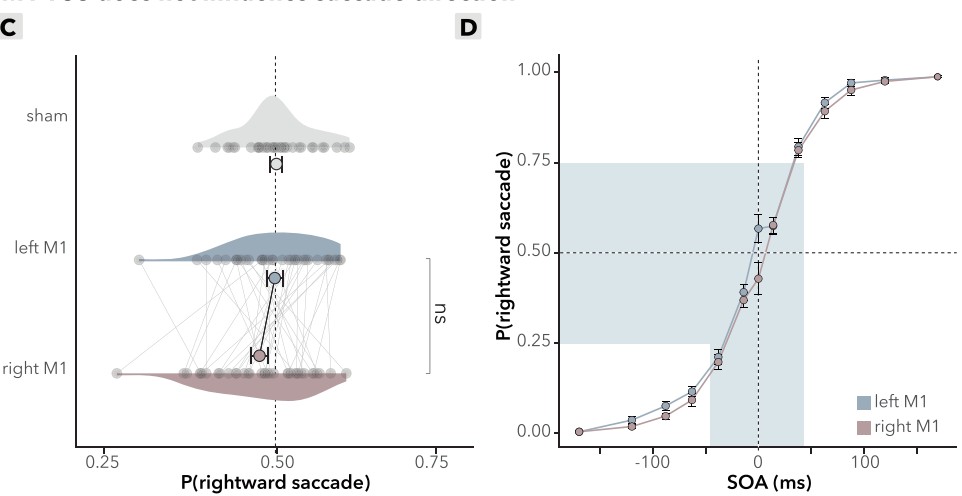

**Fig. 4 | Control analyses: M1 TUS effects. A** Individual M1 localization. BOLD responses of left and right M1 for a single subject. Participants performed a functional localizer during the intake session in which they alternated between left- and right-hand tapping movements (index finger and thumb), allowing for individual localization of the hand area in M1. **B** Acoustic simulation of TUS wave propagation. Acoustic simulations of left and right M1 for a single subject are shown. The simulation depicts the estimated intracranial intensity ($I_{sppa}$) with an intensity cutoff at the full width at half maximum (FWHM). **C** M1 TUS effects ($n = 35$). Choice-domain average effects. Gray dots represent individual participants' average saccadic direction within the choice domain. Colored dots represent group means, error bars indicate the standard error of the mean (S.E.M.) with no statistically significant group effects observed. ns $p > 0.1$. **D** M1 TUS effects ($n = 35$). Stimulation of the left and right M1 did not result in significant shifts in contralateral saccades within the choice domain (highlighted in light blue, bottom). Compared to each other, left and right M1 stimulation showed no differences in saccadic direction. Data are binned for visualization purposes into bins of approximately equal numbers of trials, resulting in SOA intervals of [0], [8.3–25.0], [33.3–50.0], [58.3–75.0], [83.3–100.0], [108.3–141.7], and [150–200]. Bins are symmetric for negative values. Note that all inferential statistics were performed using continuous SOAs. Dots represent the group mean per bin, and error bars indicate the S.E.M. across participants. **C**, **D** Source data are provided as a Source Data file. Exact $p$ values are presented in the main text. Statistical significance was determined using two-sided logistic mixed effects regressions. No multiple comparisons were applied.

In recent years, evidence has emerged for sustained and early-phase plasticity effects of TUS in humans[7,8,43]. Despite advances in in-vitro and animal models demonstrating immediate neural effects of TUS[2,4,6], the translation of these findings to humans has remained scarce[10] or potentially marred by confounds[9]. To address this, we adapted a well-established animal TUS protocol for human application, targeting the left and right FEF while participants performed a saccade choice task. This approach allowed us to assess the immediate behavioral effects of TUS on saccadic choices.

Previous lesion and brain stimulation studies have demonstrated the key role of FEF in oculomotor control, specifically mediating contralateral saccade generation[13–24]. Here, we reveal that TUS over FEFs increased the selection of contralateral saccades, particularly during with short SOAs (Fig. 2C, D). These findings, in conjunction with this past literature, suggest that the saccade bias elicited by TUS reflects a modulation of the oculomotor control by the FEF. An alternative interpretation of the results, inspired by a discussion with a reviewer, may be that the bias effects reflect changes in perception itself. The FEF has an established role in the control of visuospatial attention[17,44] and control of sensory representations in visual cortex[45,46]. Considering that attention affects temporal order

judgments[47], a complementary interpretation would be that FEF TUS altered the perceived temporal priority. Regardless, our results emphasize TUS induced changes in the sensorimotor transformations performed by the FEF in the context of eye movements.

Taken together, the directionality and context of the effect are consistent with a net facilitatory effect on FEF-mediated contralateral selection, replicating earlier work in non-human primates[2]. Note that we deliberately avoid characterizing TUS as intrinsically excitatory or inhibitory. Net effects are parameter-, physiology-, and state-dependent and reflect the local circuitry engaged at the time of stimulation: for example, a net excitatory effect could arise from either excitation of glutamatergic neurons or inhibition of GABAergic neurons. The current literature does not support a one-size-fits-all excitatory TUS recipe independent of the target region and brain state.

To demonstrate the spatial specificity of TUS, and to rule out the possibility that auditory or somatosensory confounds could drive the observed effects, we alternated blocks of FEF stimulation with blocks targeting an active control site—the hand area of the primary motor cortex (M1). Given that on a group-level, participants were able to distinguish TUS from sham trials, we highlight here that control

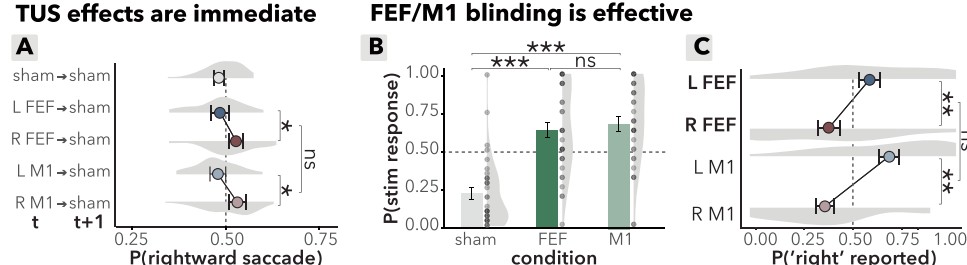

**Fig. 5 | Control analyses: interindividual variability in M1 TUS effects. A** M1 GABA+ does not predict TUS effects. Baseline GABA+ levels in the left M1 do not correlate with TUS-induced saccadic bias, calculated as the difference in probability of making a rightward saccade between left M1 TUS and sham conditions ($p = 0.5$). The line is a linear fitted group mean with a 95% confidence interval around the fit; each dot represents a participant. **B** M1 GABA+ does not predict baseline saccade behavior. Top: baseline left M1 GABA+ levels do not significantly correlate with rightward saccade probability during sham stimulation alone ($p = 1.0$). Bottom: under left M1 TUS, this correlation with rightward saccade probability remains absent ($p = 0.4$). The line is a linear fitted group mean with a 95% confidence interval around the fit; each dot represents a participant. **C** MRS voxel placement. Magnetic resonance spectroscopy (MRS) voxel placement for measuring GABA+ concentrations in the left motor cortex (M1). Color overlays represent GABA+ concentration distributions in each region. **A, B** Source data are provided as a Source Data file. Exact $p$ values are presented in the main text. Statistical significance was determined using two-sided binomial linear mixed effects regressions. No multiple comparisons were applied.

**Fig. 6 | Effects of online TUS and masking assessment. A** TUS after-effect assessment on sham trials ($n = 35$). Each sham trial was labeled based on the preceding trial's stimulation condition (e.g., "L FEF → sham" indicates a sham trial following left FEF stimulation). Density clouds represent participant distributions of saccade bias on sham trials that followed stimulation trials; dots represent group mean; error bars represent the standard error of the mean (S.E.M.). *$p < 0.05$, ns $p > 0.1$. Participants made significantly more ipsilateral saccades on sham trials following stimulation, but importantly and unlike the main TUS effect, this effect was not specific to FEF. **B** Masking, perceived stimulation (yes/no). Probability of reported stimulation perception across sham, FEF, and M1 conditions ($n = 34$). Density clouds represent participant distributions, bars indicate group means, and error bars show the S.E.M. ***$p < 0.001$, ns $p > 0.1$. Participants were significantly more likely to perceive stimulation during TUS trials compared to sham trials, highlighting that sham conditions alone may not fully account for TUS effects. No significant difference was observed between FEF and M1 conditions. **C** Masking, perceived stimulation side (left/right). Probability of reported stimulation side perception in the masking assessment ($n = 34$). Density clouds represent participant distributions, dots represent group means, and error bars show the S.E.M. **$p < 0.01$, ns $p > 0.1$. Participants were more likely to perceive TUS contralateral to the actual stimulation site. This effect was consistent across FEF and M1 regions, as indicated by the absence of a significant side-by-region interaction, supporting the robustness of the active control design. **A–C** Source data are provided as a Source Data file. Exact $p$ values are presented in the main text. Statistical significance was determined using two-sided logistic mixed effects regressions. No multiple comparisons were applied.

conditions must go beyond a generic sham and explicitly match peripheral confounds and expectancy; a consideration that is often overlooked in non-invasive brain stimulation[9,48]. Importantly, we observed no significant changes in saccadic behavior following M1 stimulation (Fig. 4C).

Our results show that TUS biases responses on trials with short SOAs, where there was high uncertainty about the correct response (Fig. 2D). On those trials, which we purposefully oversampled, we hypothesized FEF activity is more sensitive to be nudged by TUS to tip the balance in the opposite direction. This effect supports the hypothesized role of TUS in modulation of ongoing FEF computations, rather than the direct induction of saccades. TUS employs sound waves that are beyond the audible spectrum to mechanically engage with neuronal tissue, influencing characteristics such as membrane capacitance and the activity of mechanosensitive ion channels[6]. Mechanistically, unlike magnetic or electrical stimulation techniques that can directly induce neural firing (motor evoked potentials[33]; visual evoked potentials[34,35]), TUS does not directly evoke neural firing but rather modulates ongoing neural activity through subthreshold modulation and subtly nudges neural activity without inducing immediate or large-scale neural responses[36]. Thus, two principles converge here: low-evidence trials are most susceptible to small motor biases, and temporal-order judgments are sensitive to perceptual and attentional biases[17,44–47]. Together, these predict that TUS effects should peak at

short SOAs. Consistent with this, we observe the largest biases in this short SOA regime.

Finally, we established the high temporal specificity of this TUS protocol by showing it has immediate effects that occur only during stimulation trials, and do not persist into follow-up sham trials (Fig. 6A). This highlights the specificity of TUS effects to the stimulation period itself. Notably, these effects emerge very rapidly: even in trials with the fastest reaction times, where participants made saccades before the TUS duration was complete (i.e., less than 265 ms), the TUS effect was still evident and specific to the FEF. This high temporal specificity is crucial for using TUS as a tool to probe brain function with millisecond precision, aligning with principles of mental chronometry to better understand the neural dynamics underlying rapid cognitive and motor processes.

The bias toward contralateral saccades suggests a net facilitatory effect of TUS on the FEF. However, this does not necessarily imply that TUS directly excites neuronal tissue or targets specific neuron types. TUS can influence both excitatory and inhibitory neurons by altering action potential thresholds[6,49,50]. The effects of TUS can, therefore, be predicted by the baseline state of the neuronal populations involved[31,51–55]. Interestingly, our data indicate that such baseline differences in neuronal state indeed modulate the extent of TUS effects.

At baseline, participants exhibited individual biases in saccade direction, with some showing a preference for leftward and others for rightward saccades (Fig. 1B). These behavioral baselines were mirrored by neural differences, as individuals with a lower inhibitory tone, as quantified by GABA+ levels in the left FEF, tended to have a stronger intrinsic leftward bias (Fig. 3B, top). While this might appear counter-intuitive, it aligns with earlier findings from FEF MRS work[32], which demonstrated that higher GABA+ levels are associated with better suppression of distractors. In this model, cortical regions execute actions via excitatory neurons while inhibiting competing cortical regions—a process regulated by local inhibitory interneurons. Furthermore, in other motor regions, GABA+ has previously been shown to play a critical role in behavior in its relation to motor learning through action plan tuning and long-term potentiation[31,56] and FEF GABA+ predicts saccade behavior[32].

Our findings support the framework proposing that local inhibitory interneurons play a role in distraction suppression and ultimately saccade execution[32,57]. Individuals with higher GABA+ levels in the left FEF were better able to inhibit competing right FEF projections, resulting in a stronger rightward bias at baseline. Furthermore, baseline physiological inhibitory state predicted the magnitude of the TUS-induced behavioral effects, such that participants with higher cortical inhibitory tone show a larger contralateral bias under sham, and exhibit a smaller TUS-induced shift, whereas those with lower inhibitory tone exhibit a larger shift (Fig. 3A). These findings underscore the state-dependent nature of TUS and highlight the importance of considering baseline neural states when interpreting its effects. Similar to other non-invasive brain stimulation methods, the variability in TUS effects appears to be modulated by the baseline excitatory/inhibitory balance of the targeted neural populations.

Notably, the effects of TUS observed in humans are less pronounced than those documented in earlier animal work[2]. At least two reasons can be conceived. First, human FEF is larger in absolute terms, reducing the relative territory of FEF reached by the TUS focus. This might lead to a reduced efficacy of stimulation, as the average TUS intensity across the entire FEF will be lower in humans than in macaques, even with comparable peak intensities. Second, the FEF circuitry in animals appears more lateralized compared to humans[58,59], which may serve to explain the somewhat weaker effect of particularly right FEF stimulation relative to sham. A left-right asymmetry in lateralization strength is plausible given the well documented right-hemisphere dominance of human attention networks and hemispheric differences in fronto-parietal circuitry[60–63].

TUS is associated with auditory and somatosensory confounds, that could putatively drive behavioral effects masquerading as effects of TUS[9,39]. Therefore, in this study we included a number of careful controls to exclude this possibility. Importantly, the observed effects on contralateral saccades were specific to TUS over FEF, and not observed following M1 TUS. Crucially, however, confound effects were not different between these conditions, as evidenced by a masking assessment following the final session. First, while participants were able to differentiate between sham and stimulation conditions, this did not differ between M1 and FEF TUS (Fig. 6B). Second, while participants were able to distinguish left versus right TUS stimulation, again this did not differ between M1 and FEF TUS (Fig. 6C). Taken together, these potential confound effects cannot explain the main finding. Nevertheless, these observations emphasize the importance of including an active control condition rather than relying solely on a sham condition in TUS studies.

A final point of consideration is that while this study investigated baseline GABA+ levels, we did not record GABA+ as an outcome variable of TUS. Instead, our focus was on the immediate effects of TUS and its relationship to individual differences. A key direction for future research would involve assessing MRS-based changes in GABA+ to examine the potential for longer-lasting plasticity effects[8].

In summary, we demonstrated we can bias human choices using fast ultrasonic neuromodulation. Transcranial ultrasound over the FEFs, a model circuit for human decision-making, induced facilitatory effects promoting contralateral saccades. These effects were specific to FEF, immediate, present only during stimulation trials, and emerged rapidly. We showed that interindividual variability in TUS effects could be explained by the inhibitory tone of participants' FEF at baseline, with TUS modulating this balance to bring individuals into a more uniform physiological state. These findings contribute to advancing brain research by demonstrating the feasibility and efficacy of immediate TUS effects in humans. They emphasize the importance of considering interindividual variability and active control conditions in study designs. Together, this work opens new avenues for future studies aiming to explore causal brain function with high temporal precision and to develop innovative therapeutic interventions.

## Methods
### Participants
We preregistered (https://doi.org/10.17605/OSF.IO/K5P2M) a target sample size of 35 participants, based on a small to medium effect size of f – 0.35, with an alpha level of 0.05 and a power of 80% (calculated using G*Power 3.1[64]). We restricted inclusion to right-handed adults to reduce variance from hemispheric dominance and lateralization[65–71]. Participants were screened on medical history to exclude putative participants with a history of brain surgery, serious head trauma, epilepsy, convulsion, or seizure, as well as the presence of implanted metal in the head or upper body, diagnosed neurological or psychiatric disorders, and consumption of either more than four alcoholic units within the preceding 24 h or any recreational drugs within the past 48 h.

Thirty-nine participants were enrolled in the experiment. Three participants were excluded due to technical eye-tracking failure (frequent signal loss/recalibrations or inability to complete the task), and these sessions were replaced until the target sample was reached. We further excluded one participant as an accuracy outlier, defined by Tukey IQR rule (accuracy < Q1 − 1.5·IQR or >Q3 + 1.5·IQR, |z| > 3 based on the sample mean/SD, and/or robust MAD-z > 3.5) and accompanied by a non-monotone psychometric pattern—indicative of mis-understanding/non-compliance rather than sensory uncertainty (Fig. S4).

Thirty-five participants ($M_{age}$ = 24.1, $SD_{age}$ = 3.2, range = 20–32; 15:20 female:male; right-handed) were included in the final analysis. Written informed consent was obtained from all participants in

accordance with the Declaration of Helsinki, and the experimental procedures were approved by the local ethics committee (CMO2022-15953, Commissie Mensgebonden Onderzoek Oost-Nederland. Participants received €90 in total for participation, plus a performance-based bonus of up to €5 per stimulation session. Recruitment was open via university and community channels and did not restrict participation by sex, gender identity, ethnicity, or socioeconomic status. Gender identity was not systematically collected. Sex was recorded by self-report, and not included as a factor in the study design or statistical analyses because the study was not designed or powered for sex-stratified comparisons and no a priori hypotheses regarding sex differences were specified.

Saccade task results include all 35 participants. Some participants were excluded from the following analyses: For 10 participants, MRS GABA+ acquisition was of poor quality; hence, all MRS results are based on the data of 25 participants ($M_{age} = 24.7$, $SD_{age} = 3.1$, range = 20–33, 11:14 female:male). One participant did not complete the final stimulation session and was thus excluded from all masking assessment analyses; hence, all masking assessment analyses are based on 34 participants ($M_{age} = 24.9$, $SD_{age} = 3.1$, range = 20–33, 15:19 female:male). Importantly, since participants experienced all conditions in each TUS session, this participant was still included in the main analyses.

## Study overview

The study comprised three double-blind, within-subject sessions, with an interval of ~1 week (and up to 3 months) between sessions (Fig. 1F), scheduled at the same times of the day to reduce potential fluctuations in GABA+ induced by circadian rhythm. In the initial intake session, participants engaged in a practice of the saccade task without TUS delivery. Subsequently, they entered the MRI scanner to acquire structural scans. Additionally, participants completed FEF and M1 functional localizers (described below), used to target TUS during the following brain stimulation sessions. Finally, we obtained measures of GABA+ concentration from the left hemispheric stimulation regions using single voxel MRS. MRS measurements were limited to the left hemisphere due to time constraints. The left hemisphere was prioritized over the right, given the stronger lateralization in attentional processes of the right hemisphere in humans[61,72].

The two successive brain stimulation sessions incorporated a mix of sham and TUS trials during the saccade task. Each session started with screening and a 45-min preparation phase (see Neuronavigation below). Participants started with a practice block without TUS delivery or auditory masking, to reacquaint themselves with the task. For practice blocks where performance was below 60%, participants had to repeat the block. During the main task, no trials were repeated and there was no accuracy threshold. A padding block without TUS nor auditory mask bookended each TUS block. TUS transducer placement (either left and right FEF or left and right M1) was contingent on the blocks within the sequence. Sequence order was counterbalanced across participants in the two stimulation sessions (Fig. S5). TUS blocks contained three conditions: (i) left and (ii) right TUS paired with an auditory masking tone, and (iii) a sham condition with solely the auditory mask. The order of conditions within TUS blocks followed a pseudorandom pattern limited to a maximum of four consecutive trials of the same condition. At the session's conclusion, participants were queried about any symptoms they believed could be associated with TUS. This was only used for debriefing and is not further analyzes. Only after the final stimulation session, the efficacy of blinding was assessed during a short masking assessment.

## Tasks

**Saccade task.** Each trial started with fixation on a star-shaped stimulus (0.25 × 0.25 degrees of visual angle) presented at the center of the screen. After fixation, there was a delay of 300–600 ms (jittered) before the first planet-shaped target (0.5 × 0.5 degrees of visual angle, acceptance window, 3 degrees) briefly appeared in either the left or right hemifield (10 degrees of visual angle left and right from the center of the screen). Simultaneously with the appearance of this first target, TUS was delivered, lasting 500 ms. The auditory mask began 250 ms before the first target appeared and lasted 1 s, fully padding the TUS delivery. The delay between the first and second planet-shaped target ranged from 0 to 200 ms.

Target delays exhibited a non-uniform distribution, with shorter delays clustered around the central peak and the longer delays at the tails. This distribution was designed to optimize the potential for TUS-induced behavioral modulation at relatively short target delays. At the same time, it allowed to make sure that TUS does not simply induce attention lapses, characterized by incorrect responses even with long delays.

Participants were instructed to execute a saccadic eye movement to the first appearing target in either the left or right hemifield. Trial completion was followed by feedback presented for 1 s indicating indicating whether the correct target had been chosen within the designated time window (+10 points for correct responses, −10 points for incorrect responses, and 0 points for correct but late responses after 1500 ms; Fig. 1A). In this gamified task, participants could earn points and a monetary bonus of up to €5 per stimulation session based on their overall performance. During stimulation sessions, they received mixed sham and TUS trials while an auditory mask was played to blind them to the different stimulation conditions and to prevent auditory confounding. The auditory mask corresponded to the specific condition, either masking or replicating the sound of stimulation (Fig. S6).

**Functional localizers.** To prevent the risk of undershooting or missing the target due to the small ultrasound focus, we employed functional localizers to identify each participant's FEF and M1 with high fidelity[73]. The individual coordinates of interest determined using functional localizers were used for neuronavigation in the following brain stimulation sessions.

The FEF localizer[74,75] consisted of alternating 24-s blocks of saccadic eye movements and central fixation (Fig. S7A). Participants followed and fixated on a target (visual angle, 1 × 1 degrees; white square; duration, 800 ms) presented at randomized screen positions located at the left, right, or center of the screen (target distance, 14 degrees). This eye movement and fixation sequence repeated six times. Assessment of the contrast between active eye movement blocks and baseline fixation blocks allowed for localization of the left and right FEFs.

The M1 localizer[76] consisted of alternating 16-s blocks of left and right finger movement (Fig. S7B). Specifically, participants repetitively pinched their index finger and thumb together within the 16-s interval, alternating between left and right hands for six blocks per hand. This task enabled the establishment of contrasts between blocks of finger movement for each hand, providing information about left and right M1 activation.

**Masking assessment.** Following the final stimulation session, participants experienced a shorter series of sham and TUS trials involving both left and right FEF and M1. After each trial, they reported through button presses (up button for yes, down button for no) whether they believed they had received stimulation and on which side (left button for left, right button for right) they believed the stimulation was applied (Fig. S7C). The order of the three conditions was fully randomized. Additionally, the sequence of stimulation regions was counterbalanced across participants.

## Intake session

**Task practice.** Participants practiced the saccade task outside of the scanner for 15 min (198 trials) without delivery of TUS, to acquaint themselves with the task prior to the follow-up brain stimulation sessions.

**Structural and functional MRI data acquisition.** MRI scanning was performed at the Donders Centre for Cognitive Neuroimaging using a 3 Tesla Magnetom Skyra Scanner (Siemens AG, Erlangen, Germany) equipped with a 32-channel head coil. During structural scan acquisition, participants kept their eyes closed. High-resolution T1w scans were acquired (sagittal plane; repetition time (TR), 2700 ms; echo time (TE), 3.69 ms; flip angle, 9 degrees; voxel size, $0.9 \times 0.9 \times 0.9$ mm; field of view, 230 mm) for MRS voxel placement, co-registration with the functional data, and neuronavigation for TUS delivery during stimulation sessions. To capture detailed skull morphology and tissue properties for acoustic simulations of ultrasonic wave propagation, T2w scans (sagittal plane; TR, 3200 ms; TE, 408 ms; flip angle, T2 var flip angle mode; voxel size, $0.9 \times 0.9 \times 0.9$ mm; field of view, 230 mm), and UTE scans (transversal plane; TR, 3.32 ms; TE, 0.07 ms; flip angle, 2 degrees; voxel size, $0.8 \times 0.8 \times 0.8$ mm; field of view, 294 mm) were acquired.

To functionally localize the stimulation regions, a Multi-Band sequence with an acceleration factor of four (MB4) was used (TR, 995 ms; TE, 32.8 ms; flip angle, 60 degrees; voxel size, $2.5 \times 2.5 \times 2.5$ mm; field of view, $210 \times 210 \times 130$ mm acquired in axial direction). Visual stimuli of the localizer tasks were presented at the rear bore face on a flat panel screen.

**MRS data acquisition.** Magnetic Resonance (MRS) Single Voxel Spectroscopy of the left hemispheric target regions (FEF and M1) allowed for baseline GABA+ measures. For each ROI, after voxel placement based on the participant's T1-weighted scan, shimming was performed using FASTEST map[77,78] and a flip angle calibration process was carried out. For the FEF, the voxel was placed for each participant based on anatomical landmarks, at the intersect of the precentral gyrus, middle frontal gyrus and the superior frontal gyrus in the left hemisphere[11,74]. The M1 voxel was placed at the left hemispheric precentral knob located posterior to the intersection of the superior frontal sulcus that divides the superior from the middle frontal gyrus, and the precentral sulcus[79]. Baseline level of GABA+ was measured using the pulse sequence MEshcher-GArwood Point RESolved Spectroscopy (MEGA-PRESS: TR, 2000 ms; TE, 68 ms; voxel size, $2.0 \times 2.0 \times 2.0$ cm; with VAPOR water suppression[80] 128 averages and water unsuppressed reference 16 averages) as introduced by Mescher and colleagues[81,82]. The baseline level of glutamate and glutamine (Glx) was quantified using the pulse sequence Point RESolved Spectroscopy (PRESS: TR, 20000 ms; TE, 35 ms; voxel size, $2.0 \times 2.0 \times 2.0$ cm; with VAPOR water suppression 64 averages) as described by Marjańska and colleagues[83]. This data was not analyzed in the present paper.

During MRS acquisition, participants were instructed to close their eyes. MRS data acquisition started ~40 min after the end of the task practice participants completed prior to their MRI scan, thus making it unlikely that the brief task practice biased our baseline GABA + estimates[56,84–89].

### TUS sessions

**Neuronavigation and hair preparation.** The transducer was placed at the target location, and monitored throughout the session, using frameless stereotaxic neuronavigation (Localite Biomedical Visualization Systems GmbH, Sankt Augustin, Germany). We used participant specific T1w scans and x-, y-, z-coordinates of the left and right FEF and M1 derived from functional localizers. A reference tracker, five fixed markers (nasian, left and right eye, left and right ear), and 350–400 head surface markers were used to register the anatomical image to the participant's physical head. The two TUS transducers were also calibrated using a reference tracker and calibration plate. Transducers positions for the four stimulation regions were registered and quantified for acoustic ultrasonic wave propagation simulations.

Ultrasound gel (Aquaflex Ultrasound Gel, Parker Laboratories) was applied to the participant's head over stimulation regions,

followed by placement of gel pads (Aquaflex Ultrasound Gel Pad, Parker Laboratories) between the gelled head and gel-covered transducers to eliminate air bubbles[1]. Refer to Fig. S8 for a schematic set-up.

**TUS protocol.** Ultrasonic stimulation was delivered using the Neuro-FUS PRO system (Brainbox Ltd., Cardiff, UK) with two two-element ultrasound transducers (CTX250-009 and CTX250-014, 45 mm diameter, 250 kHz fundamental frequency, Sonic Concepts Inc., Bothell, WA, USA). We utilized a two-channel transducer to maximize the stimulation focal area. Although the TUS focus is characterized by a cigar-shaped profile that may extend into the white matter, it does not extend into the gray matter territory of neighboring cortical regions. The TUS protocol was adapted from Kubanek et al.[2] (pulse duration: 2 ms; pulse ramp length: 1 ms; pulse repetition frequency: 500 Hz; pulse train duration: 500 ms; square-wave equivalent $I_{sppa}$ in free water: 25 W/cm²; effective $I_{sppa}$ in free water: 9.37 W/cm²; $I_{spta}$ in free water: 9.37 W/cm²; free-field pressure: 0.9 MPa; MI: 1.76; $MI_{tc}$: 1.25; Fig. 1D, Fig. S12). Relative to the original simulation protocol in non-human primates, we used ramped pulses with an auditory mask to minimize auditory co-stimulation, increased free-water $I_{sppa}$ to offset skull attenuation, and a longer pulse train duration to match human saccade preparation and execution.

Although squared and sinusoidal ramped pulses have the same integral energy content, it is important to note that squared wave pulses have associated limitations. A squared pulse encompasses a constant intensity peak for a longer duration due to their clear onset and offset, whereas a sinus-shaped pulse exhibits a gradually increasing and decreasing peak that is never fully off. While low-intensity ultrasonic waves are beyond the range of human hearing, the on-offset of the squared pulse is detectable by humans, increasing the likelihood of auditory confounds, and thus contributing to a clearer temporal profile of stimulation[4,90]. Furthermore, since humans have a thicker skull than macaques, a higher free-field $I_{sppa}$ was applied (25 W/cm²) to match the realized intracranial intensity across species. Moreover, we adjusted the total stimulation duration to the average human saccade duration.

The temperature rise (ΔT) remained below two degrees Celsius and the derated intracranial mechanical index (MI) below 1.9 matching ITRUSST recommendations[91]. During both sham and TUS trials, an auditory mask was played through bone conducting headphones (AfterShokz, New York, US). TUS was delivered during the task through serial commands in a PsychoPy script (PsychoPy 2021.2.3[92]).

**Behavioral acquisition.** Oculomotor behavior during the saccade task was tracked using Eyelink 1000 PLUS (SR Research). Specifically, saccadic eye movements of the dominant eye were tracked from a distance of 80 cm between eye tracker and chinrest (Fig. S8A). Saccades were detected by the eye tracker's online velocity/acceleration parser using the standard cognitive-task thresholds (velocity 30°/s, acceleration 8000°/s², motion criterion of 0.15°). From the exported raw data file, we used the Python-generated event markers to identify trial boundaries and target onset, and then assigned each saccade to its corresponding trial. Note that this could result in multiple saccades per trial. Only the first qualifying saccade per trial was included in the statistical analyses. The first qualifying saccade after the target was defined by meeting the following criteria: (i) the start position within a central fixation window ($x \in [860, 1060]$ px), (ii) the endpoint within one of two lateral ROIs (Left ROI: $x \in [0, 800]$ px; Right ROI: $x \in [1120, 1920]$ px), and (iii) no blink overlapped the saccade. Direction was defined by the endpoint ROI—landings in the right ROI were labeled right and in the left ROI left. The selection criteria and raw saccadic data are illustrated in Fig. S9.

Prior to the saccade task, a nine-target calibration and validation process was conducted. Stimuli for the saccade task were programmed using PsychoPy 2021.2.3[92] and displayed on a 24-inch BenQ

monitor (resolution, 1920 × 1080; refresh rate, 120 Hz; Qisda Corporation, Taipei, Taiwan).

## Data analysis

For all regression analyses reported below, SOA was included as a z-scored covariate, and all factors were sum-to-zero coded. To account for both between and within-subject variability, saccade and masking-assessment data were analyzed with logistic mixed-effects models using the lme4 package in R[93]. We included random effects for all within subject variables. Note that for simplicity, in the equations below, the random effect structure is not included in the syntax. Furthermore, *p* values of fixed effects were acquired using Type III conditional *F*-tests with Kenward-Roger approximation for degrees of freedom, as implemented in the Anova function of the car package[94,95]. Finally, in case of significant fixed effects, post hoc pairwise comparisons were performed using the emmeans function of the emmeans package[96]. For binomial generalized linear mixed models we used simulation-based residual diagnostics (DHARMa) to verify model assumptions (Supplementary Information S.4 and Fig. S10).

**Saccade task.** Data visualization and analyses were performed using R (version 2021.9.2.382; RStudio Team, 2021). Trials on which participants made double saccades ($M = 2.1\%$, $SD = 1.6$, range = 0.4%–7.8%) and where response times exceeded 1 s ($M = 2.7\%$, $SD = 2.7$, range = 0.1–11.0%), which may have indicated failed eyetracking, were excluded. The practice and padding trials were also excluded from the dataset.

**Baseline behavior.** To evaluate the efficacy of the saccade task by establishing a robust relationship between target delay onsets and the probability of saccades to certain directions. The dependent variable is the probability of making a rightward saccade. The independent variable is target SOA. We hypothesize a higher probability of rightward saccades at larger positive SOA (e.g., target on the right hemifield appeared first) and a lower probability of rightward saccades at more negative target delays (e.g., target on the left hemifield appeared first).

The following lme4 model syntax was used:

$$saccade\ direction \sim 1 + SOA \tag{1}$$

**TUS effects.** To assess the direction of TUS effects, we looked at the effects of TUS per condition on saccade direction. We hypothesized TUS effects to surface primarily in biasing responses on trials with higher uncertainty (i.e., trials with short SOAs) and therefore focused on the "choice domain". Choice domains were defined at the individual level by determining the delay windows, i.e., SOAs, where participants showed a probability of making rightward saccades between 0.25 and 0.75 (see Supplementary Information Tables S.1–S.9 for (i) condition (all) SOA, (ii) stimulation side (left/right) by stimulation region (FEF/M1) and SOA, and (iii) MRS effect analyses for this and other choice domain window results). This led to inclusion of on average 455 trials per participant (SD = 127, range = 125–733) with an average of 65 trials per TUS condition (SD = 18, range = 15–108) and 130 trials per sham condition (SD = 37, range = 40–208).

To measure the effects of TUS on saccade behavior, we first present our primary pre-registered comparison of left versus right FEF, for which we expected opposing (lateralized) effects on saccade choices: the lateralized FEFs are each other's most informative control given their contralateral contributions to saccade selection. Next, we tested the left versus right M1 TUS effects. This step allowed us to investigate potential lateralized effects within each stimulated region. In these analyses, we again included target SOA as a continuous predictor. For example, a typical analysis model included predictors for the stimulation condition (e.g., left versus right FEF) and SOA, as follows:

$$saccade\ direction \sim 1 + condition_{leftFEF/rightFEF} + SOA \tag{2}$$

Finally, to further characterize the FEF TUS effects relative to neutral baseline conditions, we compared each FEF condition to sham:

$$saccade\ direction \sim 1 + condition_{leftFEF/rightFEF/sham} + SOA \tag{3}$$

Furthermore, to examine if the effects that we find are specific to FEF modulation, and not a result of any other confounding factors, we looked at the TUS effect of stimulation side (left vs. right) and stimulation region (FEF vs. M1) on saccade direction.

$$saccade\ direction \sim 1 + side_{left/right}*region_{FEF/M1} + SOA \tag{4}$$

Finally, in an exploratory analysis presented in Supplementary Information S.1, we examined the effects of TUS during trials where no correct choice could be made based on visual cues alone, specifically when the two targets were presented simultaneously (zero-delay trials). Here, we added the factor of zero-delay into the previous models:

$$saccade\ direction \sim 1 + condition_{leftFEF/rightFEF}*SOA_{zero/notzero} + SOA \tag{5}$$

$$saccade\ direction \sim 1 + condition_{leftM1/rightM1}*SOA_{zero/notzero} + SOA \tag{6}$$

$$saccade\ direction \sim 1 + side_{left/right}*region_{FEF/M1}*SOA_{zero/notzero} + SOA \tag{7}$$

**Biasing TUS effects.** To ascertain the specificity of TUS effects on saccade biasing rather than general perception, we investigated whether ultrasound modulates the decision curve's characteristics along different axes: horizontal shift (indicating choice bias), slope alteration (indicating impaired target discrimination), or changes in asymptotes/lapse (indicating a bias beyond the choice domain). These exploratory analyses are discussed in Supplementary Information S.2.

To estimate the slope and bias in milliseconds, we analyzed the interaction effect of condition and delay on saccade direction, focusing specifically on a delay range of −75 to +75 ms to increase sensitivity for detecting any slope effects. This range was selected because it closely approximates the individual choice domain used in other analyses, ensuring consistency and comparability across methods. Unlike previous analyses where individual choice domains were used, we opted for a fixed delay range in this analysis. This decision was made because we aimed to quantify the absolute value of the bias shift (horizontal shift of the curve) in milliseconds. Using the scaled individual choice domains does not provide the opportunity to calculate this fixed bias shift in absolute time units. By including condition and delay as random effects, we were able to estimate the random slopes and biases for each participant. This approach allowed us to determine whether TUS induced horizontal shifts in the decision curve, indicative of a choice bias.

$$saccade\ direction \sim 1 + condition_{sham/leftFEF/rightFEF/leftM1/rightM1}*SOA \tag{8}$$

For lapse estimation, we aimed to understand whether TUS could evoke saccades at longer delays, thus indicating an effect beyond mere biasing. We selected absolute delays between 75 and 200 ms and assessed whether choice accuracy depended on condition and absolute delay. By focusing on absolute delays, rather than distinguishing between negative and positive delays, we prioritized analyzing overall accuracy rather than side-specific biases. This choice was made because we do not expect side biases to play a role in this context; instead, we are interested in understanding general task performance and accuracy. Therefore, this analysis focused on the asymptotes of the decision curve to determine if TUS influenced saccade behavior

even when the delays were long, and the task was easy.

$$saccade\ accuracy \sim 1 + condition_{sham/leftFEF/rightFEF/leftM1/rightM1} + SOA_{abs}$$
$$(9)$$

**Online TUS effects.** Moreover, to assess whether this protocol truly functions as a short-lived protocol without producing longer-lasting effects, each sham trial was labeled according to the preceding trial (e.g., left_FEF–sham refers to a sham trial that followed a left FEF trial). However, note that this analysis was conducted with a mean of only 22 trials per condition ($SD = 7$, range = 6–43). We then ran the same side-by-region model to analyze these labeled sham trials. Given the expectation that the protocol only exerts direct, immediate effects, we hypothesized that there would be no significant interaction effect observed. Furthermore, we also performed a Bayesian ANOVA using the same model syntax, as this approach provides a more robust assessment of evidence for the null hypothesis.

$$saccade\ direction \sim 1 + side_{sham(left/right)}{}^{*}region_{sham(FEF/M1)} + SOA \quad (10)$$

**Functional localizers.** To accurately target the stimulation sites for each individual, the participants performed a FEF and M1 localizer during the intake session. This data was then pre-processed and analyzed to obtain coordinates for each region per participant.

fMRI pre-processing and analysis were conducted using SPM12 in MATLAB R2023a, along with MRIcroGL for result visualization. The initial steps included excluding the first five fMRI volumes to account for signal steady-state transition, converting IMA files to DICOM compatible format, and visually checking for artefacts. We performed both single subject and group level analyses ($N = 35$) to establish coordinates within native and standard space, respectively (FEF: Fig. 2A (single-subject) and 3C (group-level); M1: Fig. 4A (single-subject) and 5C (group-level)).

Realignment and reslicing were performed for both levels, followed by coregistration with the participant's T1w-image for single subject analysis and with Montreal Neurological Institute standard space for group level analysis. Data was smoothing with a 6 mm FWHM Gaussian kernel, and realignment parameters were inspected. The blocks were convolved with canonical hemodynamic response function, followed by voxel-wise fitting of a general linear model, resulting in the computation of statistical parametric maps for the comparisons. Subsequently, beta weights for each condition were estimated to create contrast maps, enabling Family-Wise Error corrected cluster-level inferences ($p < 0.05$). For the FEF localizer, saccades minus fixation blocks were used to obtain coordinates for the left and right FEF. The M1 localizer contrasts involved pinching blocks of right fingers minus left fingers and vice versa to identify the left and right M1, respectively.

To determine FEF targets for TUS delivery, we selected a peak voxel within the significant cluster within FEF, specifically at the junction of the superior precentral sulcus and the superior frontal sulcus. The FEF localizer required reflexive pro-saccades, activating both medial and lateral FEF peaks. The medial peak, linked to higher order cognitive control[75,97,98], was selected for TUS targets. This decision aligns with our hypothesis that TUS holds the highest potential for influencing saccadic behavior at equal preference, requiring the execution of voluntary saccadic eye movements by FEF. The M1 localizer with pinching either the left or right finger elicited a distinctive activation cluster of significant voxels in both left and right M1. Within the activation cluster, the local maximum of peak voxel was selected for the x-, y-, and z-coordinates.

The accuracy of selected coordinates within sulci branches was assessed with FSLeyes by means of visualizing effect sizes modulated by statistical significance with transparent threshold. Once confirmed, established coordinates per stimulation region were entered in the

Localite software to plan and monitor TUS delivery. Group level analysis calculated contrast estimates' standard error and mean, determining significance of the average estimate.

**MRS analysis.** To investigate interindividual differences in TUS susceptibility, we quantified baseline inhibitory tone in the left hemisphere stimulation sites (left FEF and left M1) using MRS.

GABA+ concentrations were quantified using Gannet version 3.1.4[99], with water used as a reference. Gannet's standard preprocessing pipeline was used, which includes frequency and phase correction by spectral registration and line broadening. Edited spectra were generated by subtracting individual edit-ON spectra from edit-OFF spectra. Notably, the editing approach not only targets GABA but also other macromolecules at 3 ppm, therefore the concentrations of GABA + (GABA and macromolecules) are reported. Gray matter, white matter and CSF tissue fractions for determining tissue-corrected concentrations were obtained for both voxels using SPM12. Metabolite concentrations were then relaxation and tissue-corrected (Gasparovic et al. method)[100].

To ensure data quality, two independent researchers performed visual quality checks of the data. Using the GannetLoad output, water frequency drift was assessed to identify excessive movement artifacts. Next, Cr signal alignment was inspected to evaluate the quality of frequency alignment. For participants with noticeable drift or misalignment, the affected averages were removed, and GABA+ quantification was reprocessed using Gannet. Participants were excluded if more than 50% of their averages had to be removed or if drift and alignment remained insufficient despite reprocessing, as their GABA+ was unreliable or inestimable due to lipid contamination or low signal-to-noise ratio (SNR). Reliable model fits were achieved for 25 out of 35 acquisitions. Data quality was further quantified using the SNR and full-width-at-half-max (FWHM) of N-acetylaspartate (NAA) and fit error of the GABA+ peak provided by Gannet (Fig. S11A–C).

To assess whether interindividual variability in saccade bias could be explained by baseline inhibitory tone, we examined whether the probability of making a rightward saccade in the left FEF and sham conditions (as well as in the control left M1 and sham conditions) was influenced by baseline GABA+ levels. Given that we only measured the left hemispheric target regions using MRS, we restricted our analyses to the left hemisphere. Specifically, we tested whether the interaction between condition (left FEF vs. sham) and baseline GABA+ levels predicted saccade direction, with target delay included as a separate predictor. We ran the same model for M1 GABA+, to assess if M1 GABA+ levels were predictive of the M1 TUS effects or intrinsic bias.

$$saccade\ direction \sim 1 + condition_{sham/leftFEF}{}^{*}FEF_{GABA+} + SOA \quad (11)$$

$$saccade\ direction \sim 1 + condition_{sham/leftM1}{}^{*}M1_{GABA+} + SOA \quad (12)$$

**Masking assessment.** To evaluate the efficacy of participant blinding to different conditions, we investigated whether participants could distinguish sham from TUS trials by analyzing if stimulation perception (yes/no) depended on the stimulation condition (sham/FEF/M1). Additionally, we assessed whether stimulation and side perception differed between FEF and M1 conditions. Specifically, we analyzed if side perception (left/right) depended on the stimulation side (left/right) and region (FEF/M1) in the following mixed models:

$$stimulation\ perception \sim 1 + condition_{sham/FEF/M1} \quad (13)$$

$$side\ perception \sim 1 + side_{left/right}{}^{*}region_{FEF/M1} \quad (14)$$

For both models, we also performed a Bayesian ANOVA to further assess the evidence for the null hypothesis. We hypothesized that

there would be no difference in stimulation perception between sham, FEF, and M1 conditions due to the delivery of the auditory mask. Even if differences in stimulation perception were found compared to sham, we expected this not to be problematic due to the inclusion of the active control site (M1), where TUS was also delivered to both the left and right hemispheres. We anticipated no significant differences between FEF and M1 under these conditions. While differing results in side perception might be observed, this would not pose a problem since M1 stimulation is also lateralized.

### Reporting summary

Further information on research design is available in the Nature Portfolio Reporting Summary linked to this article.

## Data availability

The anonymized behavioral data generated in this study (trial-level tables), task timing logs, stimulation targets/coordinates, single-subject localizer peaks, and summary spectroscopy outputs with QC (i.e., the dataset required to reproduce the results reported in this paper) have been deposited in the Radboud University Data Repository under https://doi.org/10.34973/drtg-kq58. The raw MRI data are protected and are not available due to data privacy laws (GDPR) and the study's ethics approval (CMO Oost-Nederland, CMO2022-15953). Source data are provided with this paper.

## Code availability

All behavioral and fMRI task code and all analysis scripts are available at https://doi.org/10.34973/drtg-kq58.

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

## Acknowledgements

This experiment was supported by the Dutch Research Council (NWO), awarding VIDI fellowships to L.V. (18919) and H.E.M.d.O. (452-17-016). We would like to acknowledge Edward J. Auerbach, Ph.D., and Małgorzata Marjańska, Ph.D. (Center for Magnetic Resonance Research and Department of Radiology, University of Minnesota, USA) for the development of the pulse sequences for the Siemens platform, which were provided by the University of Minnesota under a C2P agreement. Additionally, we thank Norbert Hermesdorf, Margely Cornelissen, Hubert Voogd, Sibrecht Bouwstra, Gerard van Oijen, and Pascal de Water from the technical support group at the Donders Centre for Cognition, Faculty of Social Sciences, Radboud University, for their excellent technical assistance and support throughout this study. Finally, we would also like to thank Marwan Engels (Donders Centre for Cognition, Radboud University) for his substantial role during data acquisition, and Sjoerd Meijer (Donders Centre for Cognitive Neuroimaging, Radboud University) for his contribution to preparing the ethics documentation and to setting up the laboratory infrastructure for this study.

## Author contributions

S.F., L.V and H.E.M.d.O. conceptualized and designed the experiment; A.C., S.L.Y.W. and S.F. designed and programmed the behavioral and functional localizer tasks; S.F. and S.L.Y.W. collected the data; J.P.M. set up the fMRI and MRS sequences; S.F., L.V., and H.E.M.d.O. analyzed the behavioral, functional and spectroscopy data; B.R.K. contributed to behavioral data analysis; R.S.K. contributed to spectroscopy analysis; S.L.Y.W. contributed to fMRI analysis; S.F., L.V and H.E.M.d.O. wrote the manuscript; B.R.K., R.S.K., W.P.M., J.P.M. and A.C. revised the manuscript.

## Competing interests

L.V. declares no competing interests relevant to this study. L.V. is a board member of ITRUSST and the Brainbox Initiative. L.V. has received non-financial support from Image Guided Therapy SA (France), Sonic Concepts LLC (US), and Brainbox Ltd (UK), and consulting fees from Nudge LLC (US). All other authors declare no competing interests.
