## [Transparent Peer Review file · Nature Communications]

Rapid modulation of choice behavior by ultrasound on the human frontal eye fields

Corresponding Author: Ms Soha Farboud

Version 0:

Reviewer comments:

Reviewer #1

(Remarks to the Author)

This manuscript reports an interesting study in a group of 35 human participants in a saccade task for behavior assessment and MRS imaging. Transcranial ultrasound was delivered at the frontal eye fields (FEF), and induced robust facilitatory effects promoting contralateral saccades, which is immediate, present only during stimulation trials, and emerged rapidly. Individual subject TUS effect sizes were found to be correlated with baseline FEF GABA+ levels. Despite relatively rich literature in animal studies, translation of TUS to human applications remains unclear. This work thus makes an interesting contribution to better understanding of human TUS research. There are however some comments that need to be addressed.

1. Focusing analysis on trials with <75% accuracy seems a bit strange. The authors justify this choice by stating that TUS is expected to nudge brain function in one direction or another when sensory evidence is low, but including trials <50% accuracy would indicate the participants are effectively operating at random chance or lower. Wouldn't focusing analysis on a range of ~60% (higher than random chance) - ~90% (less than 100% certainty) be more indicative of low sensory evidence that is still biologically meaningful towards the intended outcome? After reading the Methods, it seems that all trials with less than 60% accuracy were redone until the accuracy was above 60%. This should be mentioned when discussing the included accuracy range.
2. Throughout their various figures, the authors bin their data into intervals of 0 to 1, 1 to 26, 26 to 50, 50 to 75, 75 to 100, 100 to 142, and 142 to 200 ms. Can they provide a bit of rationale for these unequal bin sizes?
3. Authors present a statistically significant difference in saccade probability for left vs. right FEF stimulation in Figure 2. What about comparison to the sham condition?
4. The authors excluded subjects due to poor eye-tracking ability and low accuracy in the saccade task (<60%). This seems contradictory to their previous rationale for including trials with low sensory information. Wouldn't these subjects be excellent candidates for exploring TUS' ability to nudge uncertain biological signals to a meaningful outcome?
5. It is mentioned in the Methods that all subjects were right-handed, indicating perhaps this was an intentional choice? If that is the case, it may be worth including a line or two about why this choice was made.
6. The subjects practiced the saccade task for 15 minutes immediately prior to their MRI. Would this influence the recorded GABA+ levels?
7. Authors report Isppa and MI. What about the free field pressure itself? Ispta?
8. The authors, in the discussion, claim "TUS effects are dependent on individual baseline GABA+ levels." Is it appropriate to assert causality from the included data? Would it be more accurate to say they are predicted by or correlated with?
9. The authors performed their analysis with linear mixed effect models. These models have certain underlying assumptions about data structure (i.e., normality of residuals, homoscedasticity, linearity, etc). Were these assumptions checked?

10. The linear mixed effect models used to analyze each metric are somewhat complex, often times with several fixed and random effects. Readers may benefit from a bit more discussion on what the equations mean in addition to the provided syntax.

11. For their primary analysis, it seems the authors made multiple models per each condition (for example: line 643 has a model specifically for left FEF vs. right FEF), and then analyzed each model separately for significance. However, in the supplementary analysis (line 689), they show a model including all FEF and M1 conditions together. The latter seems more intuitive for analysis, as it would directly allow for the emmeans pair-wise comparisons between all FEF, all M1, and all sham conditions. Why did they authors choose to not include all conditions in the same model(s) for their primary analysis?

12. The authors made many statistical comparisons, but did not seem to apply any kind of p-value multiple comparison correction. Given that multiple comparison correction seems to be the TUS field standard, they may want to elaborate on why they chose not to do so.

13. The authors present right M1 vs. left M1 behavioral comparisons in Figure 4, separately from the right FEF vs. left FEF. It may help strengthen their results if they can combine these results into a single figure to demonstrate that FEF stimulation was not only different from each other, but also different from the results of their spatial control.

14. One of the main claims in the Summary is that their work “highlight[s] the importance of neurophysiological state in neuromodulation.” However, in their discussion, they write, “TUS acts to normalize interindividual differences in physiological states.” Are these two statements contradictory? If TUS is normalizing the interindividual effects, wouldn't the baseline neurophysiological state be unimportant?

15. The authors only had time to collect one hemisphere's worth of GABA+ readings, so they are unable to see if baseline right FEF GABA+ is correlated with right TUS effect sizes. However, as an additional control, could they investigate whether left FEF GABA+ is correlated with right TUS effects?

(Remarks on code availability)

Reviewer #2

(Remarks to the Author)

(Remarks on code availability)

Reviewer #3

(Remarks to the Author)

The authors develop the online TUS technique that allows trial-based comparisons within blocks. Although not highlighted sufficiently, in my opinion, it is quite interesting that various TUS after-effects can be elicited by auditory/somatosensory confounds depending on the experimental settings, which emphasizes the importance of appropriate control of experimental procedures. I would raise several points to be addressed that I hope will improve this manuscript.

The stimulation pulse was structured, with fundamental frequency of 250 kHz, ramping up and down sinusoidally within 2 msec (up: 1 msec, down: 1 msec), and lasting for 500 msec, if I understand correctly. I think the excitatory effect of online TUS is rare and wondered which component of the structured stimulation pulse elicited excitatory effect on the FEF. It would also be interesting to see historical summary of how the stimulation protocol was developed.

If it is possible to estimate, based on the eye movement results, how robustly the stimulation protocol can modulate behavior in cognitive tasks, that would be helpful.

It would be informative to show raw data of eye movements and how saccade direction was determined.

Three hypothetical TUS effects (excitatory, inhibitory, and perturbatory shown in Fig. 1E) have distinct patterns, but unfortunately the curve for the sham condition was not shown in Fig. 2D.

Was the mysterious biased saccade direction around zero SOA after M1 stimulation observed in previous studies of this kind?

The saccade direction in sham trials following sham trials appeared biased toward the left side (Fig6A). If the left side bias is a general natural tendency, an alternative interpretation for the after-effect assessment results would be possible.

(Remarks on code availability)

Reviewer #4

(Remarks to the Author)

Farboud et al describe experiments in which they test the hypothesis that (relatively) short (500 ms) transcranial ultrasound (TUS) pulses delivered to the frontal eye field (FEF) in humans should bias saccadic eye movements as they appear to in nonhuman primates (e.g. ref. 2). The authors point out that the large majority of human studies to date focus on repetitive 'offline' protocols with temporally sustained effects of TUS, while those in animals aim for short effects, which provide great utility in addressing more circuit-level effects. Using the short protocol, the authors indeed report that TUS delivered the FEF (left and right) increase the probability of contralateral saccades, like results in nonhuman primates. In addition, the authors measured GABA levels in the (left) FEF via magnetic resonance spectroscopy (MRS) and found that the level of GABA predict the bias in saccades as well as the effects of TUS. Overall, the results are relatively straightforward and fairly convincing. I only have a few comments/suggestions that I believe would improve the interpretation and soundness of the results.

Major comments.

1. The authors aim to emulate a previous study of ultrasound in nonhuman primates (ref. 2), yet the task employed differs in an important way. In the previous study (and others), subjects perform a free-choice task in which they can choose either target. However, the onset asynchrony typically causes them to choose the first target. Nonetheless, subjects (monkeys or humans) are not instructed to saccade to the first target, as in this study. While I do not think this is a critical issue, I do think it exposes an important point regarding the interpretation of the present results. In the free-choice task, one might argue that biases in saccades elicited by stimulation are more related to motor control and motor decisions. In contrast, when the task is explicitly a perceptual one, i.e. 'saccade to the one perceived as appearing first' one might argue that the bias effects are more likely to reflect changes in perception. The distinction, while not crucial here, is important given that the FEF has an established role in the control of visuospatial attention in both human and nonhuman primates (e.g. <https://www.pnas.org/doi/abs/10.1073/pnas.98.3.1273>; <https://ieeexplore.ieee.org/abstract/document/6789738>) and has been shown to control sensory representations in visual cortex (e.g. <https://www.nature.com/articles/nature01341>; [https://www.cell.com/current-biology/fulltext/S0960-9822\(06\)01818-5?cc=y%3Fcc=y](https://www.cell.com/current-biology/fulltext/S0960-9822(06)01818-5?cc=y%3Fcc=y)). Thus, an alternative interpretation of the effects of TUS on FEF observed would be that subjects perceived the onset in the contralateral hemifield as earlier than without TUS due to attentional deployment. Since it has been shown that attention affects temporal order judgments (e.g. <https://psycnet.apa.org/fulltext/1991-26439-001.html>), this interpretation is entirely sound. Moreover, the effects would be expected to be more robust within the so-called 'choice-domain' that the authors focus on (Figure 2D).

Again, I don't think this bears critically on the importance of the results described, but they authors definitely need to modify/expand their conclusions to more accurately reflect what is known about the FEF's role in visually guided behavior.

2. The differences in magnitude and variability of TUS effects on saccades shown in figure 2C appear to be noteworthy. Although the authors do briefly refer to differences in lateralization of function between humans and animals (discussion) they don't appear to make much of the apparent differences between left and right FEF TUS (e.g. <https://www.sciencedirect.com/science/article/pii/S2352250X18302392>). Perhaps it would be worth addresses this in their interpretation of the results and the extent to which their results are consistent with past evidence.

(Remarks on code availability)

Version 1:

Reviewer comments:

Reviewer #1

(Remarks to the Author)

Farboud et al have addressed the raised questions thoughtfully. Their manuscript is well written, and their figures are informative. Their results are straightforward, and consistent with prior work done in nonhuman primates.

My remaining concern is with the strength of the data relative to the control condition(s). The authors have now provided a direct pairwise comparison between the audio mask sham condition and the FEF-targeted tFUS conditions. However, these direct comparisons are not particularly favorable ($p < 0.1$ for right FEF vs. sham).

Similarly, for the now-provided comparison on FEF and M1 targeted tFUS model, the Cohen's d effect size is only 0.13. I think the usual interpretations for Cohen's d values consider weak effects ≥ 0.2 . Less than 0.2 is, to my knowledge, very tiny or even negligible.

Given that the direct comparisons between the FEF targeted tFUS and the control conditions are not all that strong, perhaps the authors should tone down their language throughout the manuscript (they refer to their findings as "robust" numerous times).

One minor thing: Figure S12 – Should spatial-peak pressure amplitudes be 0.9 MPa, not kPa, based on the written description?

(Remarks on code availability)

Reviewer #2

(Remarks to the Author)

(Remarks on code availability)

Reviewer #3

(Remarks to the Author)

The authors addressed all my points raised in the previous version of the manuscript satisfactorily.

(Remarks on code availability)

Rapid modulation of choice behavior by ultrasound on the human frontal eye fields

Soha Farboud, Benjamin R. Kop, Renée S. Koolschijn, Solenn L.Y. Walstra, José P. Marques, Andrey Chetverikov, W. Pieter Medendorp, Lennart Verhagen, Hanneke E.M. den Ouden

REVIEWER COMMENTS

We thank the reviewers for their thoughtful comments. In our reply, we use the following font styles for clarity:

- reviewer text in blue,
- reply to reviewers in normal font,
- *"manuscript text in italics and orange"*
- ***"changes in the manuscript text in bold, italics and orange"***

Reviewer #1 (Remarks to the Author):

This manuscript reports an interesting study in a group of 35 human participants in a saccade task for behavior assessment and MRS imaging. Transcranial ultrasound was delivered at the frontal eye fields (FEF), and induced robust facilitatory effects promoting contralateral saccades, which is immediate, present only during stimulation trials, and emerged rapidly. Individual subject TUS effect sizes were found to be correlated with baseline FEF GABA+ levels. Despite relatively rich literature in animal studies, translation of TUS to human applications remains unclear. This work thus makes an interesting contribution to better understanding of human TUS research. There are however some comments that need to be addressed.

1. Focusing analysis on trials with <75% accuracy seems a bit strange. The authors justify this choice by stating that TUS is expected to nudge brain function in one direction or another when sensory evidence is low, but including trials <50% accuracy would indicate the participants are effectively operating at random chance or lower. Wouldn't focusing analysis on a range of ~60% (higher than random chance) - ~90% (less than 100% certainty) be more indicative of low sensory evidence that is still biologically meaningful towards the intended outcome? After reading the Methods, it seems that all trials with less than 60% accuracy were redone until the accuracy was above 60%. This should be mentioned when discussing the included accuracy range.

We appreciate the suggestion and have clarified our rationale for focusing on the choice domain trials. Unlike TMS or DBS, which can directly elicit evoked potentials, TUS primarily modulates ongoing activity in a subthreshold manner (Butler et al., 2022; Yoo et al., 2022). Consequently, any behavioral effect is expected to be most evident when intrinsic signals are weak. This phenomenon has been first described in humans in the context of a dot-motion-task, where the experimenters specifically designed the task to oversample trials with ambiguous stimuli (Butler et al., 2022). In the case of the current study, the sensory evidence is more ambiguous when there is a short SOA. We therefore designed the task to oversample these short-SOA trials (Figure 1C). To be sensitive to individual differences in accuracy and sensory ambiguity, we titrated the range to the individual level of the SOA

range where participants were < 75% correct. This threshold explicitly considers a symmetric range whereby 25-75% of the choices are rightward, including the 50% equivalence point when sensory evidence alone is not enough to make a correct choice. In summary, we agree with the reviewer's notion of titrating the accuracy to difficult trials. However, whereas the reviewer suggests a range of 60-90%, based on previous literature (Butler et al., 2022; Kubanek et al., 2020), we have deliberately chosen a more ambiguous range that explicitly includes the sensorimotor equivalence point. This represents decisions under high uncertainty / low sensory evidence, where a small perturbation has the potential to bias the outcome. For performance on trials at very short SOAs, moment-to-moment sensory noise brings the decision variable close to threshold, and TUS may tip the balance. Note that (near) chance level performance is exactly what we would expect for trials with a very short SOA and is thus not concerning. In fact, excluding trials below 60% would remove the most sensitive portion of the psychometric function where TUS could exert its effects. We now further clarify this point in the Results and Methods:

In Results, §2.1 (pages 6 - 7, lines 106-118):

"In contrast to magnetic or high-voltage electrical stimulation, which can directly elicit evoked responses (motor-evoked potentials³³; visual-evoked potentials^{34,35}), TUS at the low intensities employed in the current study is thought to act through modulating sub-firing threshold, ongoing activity, thereby gently 'nudging' network dynamics rather than producing immediate, large-scale neural responses³⁶. Therefore, we expected TUS effects to bias responses primarily on trials with short SOAs, i.e. high uncertainty about which target appeared first. On these trials, with low sensory evidence for left- or rightward responses, a small perturbation has the potential to bias the outcome, and thus we expected that TUS could 'tip the scales'². In contrast, for long SOAs there was low uncertainty about the correct response and TUS would not be able to overrule the associated robust neural signal. Therefore, we designed the task to oversample trials with shorter SOAs (Figure 1C) and focused the primary analysis on trials with SOAs where participants were <75% correct (Figure 1B; hereafter referred to as the choice domain)."

Furthermore, in the Supplementary Tables S1-9 we confirm that the excitatory FEF-TUS effect does not depend on the exact 25-75% choice-domain definition. Using wider participant-specific windows (20-80% and 15-85%), the direction and magnitude of the TUS-induced bias remain stable (with the expected attenuation as higher-evidence trials are included).

Finally, the <60% criterion for trial repetition applied only to the brief practice block at the start of each session to ensure participants re-familiarized themselves with the task; practice trials were not analyzed. We have clarified this in the Methods:

In Methods, under "Study overview" (page 31, lines 502-504):

"Participants started with a practice block without TUS delivery or auditory masking, to reacquaint themselves with the task. For practice blocks where performance was below

60%, participants had to repeat the block. During the main task, no trials were repeated, and there was no accuracy threshold."

2. Throughout their various figures, the authors bin their data into intervals of 0 to 1, 1 to 26, 26 to 50, 50 to 75, 75 to 100, 100 to 142, and 142 to 200 ms. Can they provide a bit of rationale for these unequal bin sizes?

Apologies for the lack of clarity. First, binning is used only for visualization; all statistical models use continuous SOA values. Because short SOAs were oversampled by design, in the figures we used unequal bin widths to yield comparable trial counts per bin and to provide higher visual resolution around threshold. We further realized that we made a mistake in the description, as the reported bin sizes suggested a higher temporal resolution than the screen refreshment rate of 120 Hz. We now adjusted the figure legends to clarify:

Figure 1B, 2D and 4D caption (pages 8, 11, 13):

"Data are binned for visualization purposes into bin sizes of approximately equal numbers of trials, resulting in SOA intervals of [0], [8.3 - 25.0], [33.3 - 50.0], [58.3 - 75.0], [83.3 - 100.0], [108.3 - 141.7], and [150 - 200]. Bins are symmetric for negative SOA values. Note that all inferential statistics were performed using continuous SOAs."

3. Authors present a statistically significant difference in saccade probability for left vs. right FEF stimulation in Figure 2. What about comparison to the sham condition?

Our preregistered primary hypothesis targeted the left-vs-right FEF contrast, as the two FEFs serve as each other's most informative control condition, given their contralateral contributions to saccade selection. We therefore present this as our primary analysis. The sham condition is a neutral baseline, expected to lie between the two FEF conditions; accordingly, FEF-vs-sham pairwise differences are expected to be smaller. However, as we understand that this might still be of interest, we now also display the FEF-vs-sham comparisons in Figure 2C (see below) and report this in the Results and Methods. In this comparison, the sham condition does indeed fall between the left and right FEF conditions, but the difference from neutral sham is statistically stronger in one hemisphere: left FEF vs sham is significant, whereas right FEF vs sham is a trend. We added a discussion regarding this observation (also suggested by Reviewer 4).

In Methods, under "TUS effects" (page 41, lines 718-729):

"To measure the effects of TUS on saccade behavior, we first present our primary pre-registered comparison of left versus right FEF: the lateralized FEFs are each other's most informative control given their contralateral contributions to saccade selection. Next, we tested the left versus right M1 TUS effects. This step allowed us to investigate potential lateralized effects within each stimulated region. In these analyses, we again included target SOA as a continuous predictor. Finally, to further characterize the FEF TUS effects relative to neutral baseline conditions, we compared each FEF condition to sham."

In Results, §2.2 (page 9, lines 130-134):

“This excitatory behavioral effect on contralateral saccades was not observed for stimulation to left versus right M1 (details reported below; for visual comparison of FEF-sham and FEF-M1 refer to Figure S1). Comparison of each FEF condition relative to baseline (sham) showed that the sham condition provided a baseline between left and right FEF TUS effects, whereby left FEF TUS increased the proportion of rightward saccades ($b = -0.12$, $p = 0.023$), and right FEF showed a trend toward more leftward saccades ($b = 0.12$, $p = 0.08$).”

In Figure 2C (page 10):

(C) FEF TUS effects. Choice-domain average effects. Grey dots represent individual participants' mean saccadic directions within the choice domain. Colored dots represent group means, error bars indicate the standard error of the mean (S.E.M.). *** $p < 0.001$, * $p < 0.05$, † $p < 0.1$.

In Discussion (pages 23-24, lines 366-373):

“Notably, the effects of TUS observed in humans are less pronounced than those documented in earlier animal work². **At least two reasons** can be conceived. First, human FEF is larger in absolute terms, reducing the relative volume of FEF reached by the TUS focus. This might lead to a reduced efficacy of stimulation, as the average TUS intensity across the entire FEF will be lower in humans than in macaques, even with comparable peak intensities. **Second, compared the FEF circuitry in animals is more lateralized compared to humans^{58,59}, which may lead to weaker effects of particularly right FEF stimulation. A left-right asymmetry is plausible given the well-documented right-hemisphere dominance of human attention networks and hemispheric differences in fronto-parietal circuitry⁶⁰⁻⁶³.**”

4. The authors excluded subjects due to poor eye-tracking ability and low accuracy in the saccade task (<60%). This seems contradictory to their previous rationale for including trials with low sensory information. Wouldn't these subjects be excellent candidates for exploring TUS' ability to nudge uncertain biological signals to a meaningful outcome?

Our original phrasing was unclear. The exclusion criteria were designed to protect data quality, not to remove low-evidence trials. We distinguish trial-level uncertainty (very small SOAs within otherwise well-performed sessions), which is central to our hypothesis and retained, from participant-level unreliability, which compromises internal validity.

Concretely, one participant was excluded as an accuracy outlier across the main task, consistent with non-compliance. This participant displayed low accuracy across the range, also in the SOA domain where other participants approximated 100% accuracy. In addition, three participants were excluded for eye-tracking instability/technical failure (frequent signal loss/recalibrations), rendering gaze data unreliable and unrelated to TUS. These cases were identified and replaced during data collection, allowing the target sample of $N = 35$ to be reached. We have clarified this in the Methods and added Supplementary Figure S4 illustrating the behavior of the excluded participant compared to the rest of the sample.

In Methods, under "Participants" (page 29, lines 461-468):

"Thirty-nine participants were enrolled. Three participants were excluded due to technical eye-tracking failure (frequent signal loss/recalibrations or inability to complete the task), and these sessions were replaced until the preregistered target sample was reached. We further excluded one participant as an accuracy outlier, defined by Tukey IQR rule (accuracy < $Q1 - 1.5 \cdot IQR$ or > $Q3 + 1.5 \cdot IQR$), $|z| > 3$ based on the sample mean/SD, and/or robust MAD- $z > 3.5$) and accompanied by a non-monotone psychometric pattern—indicative of misunderstanding/non-compliance rather than sensory uncertainty (Figure S4)."

Figure S4 | Participant accuracy and psychometric quality checks

(A) Distribution of overall accuracy across participants. The shaded curve is a kernel-density estimate; the horizontal boxplot shows the median (line), quartiles (box), and whiskers. Participants were flagged as outliers if they met any robust criterion: (i) Tukey IQR rule (accuracy $< Q1 - 1.5 \cdot IQR$ or $> Q3 + 1.5 \cdot IQR$), (ii) $|z| > 3$ based on the sample mean/SD, and/or (iii) robust MAD-z > 3.5 . The red marker highlights sub-036 that exceeded all three criteria and is flagged as an outlier. (B) Empirical psychometric curves plotting the probability of a rightward saccade against SOAs (ms; rightward-positive), collapsed across all conditions. Light grey ribbons indicate the

S.E.M. across trials per SOA bin. Most participants show the expected monotonic increase from leftward to rightward delays (horizontal dashed line at 0.5 = chance, and vertical dashed line at 0 SOA). In contrast, sub-036 (red panel) exhibits non-monotone, noisy behavior with extended near-chance performance, consistent with non-compliance/unstable responding.

5. It is mentioned in the Methods that all subjects were right-handed, indicating perhaps this was an intentional choice? If that is the case, it may be worth including a line or two about why this choice was made.

This was indeed intentional. We restricted inclusion to right-handers to reduce between-subject variance in the hemispheric lateralization of FEF and M1. Human attention/saccade control shows right-hemisphere dominance at the population level (Anderson et al., 2012; Shulman et al., 2010; Vossel et al., 2014). Mixing handedness groups would increase heterogeneity in these lateralized systems and could obscure or mimic hemisphere-specific TUS effects. Second, our M1 control condition benefits from a homogeneous motor organization: handedness is associated with differences in motor network lateralization/effective connectivity (Johnstone et al., 2021; Pool et al., 2014; Tomasi & Volkow, 2024; see also our answer regarding FEF lateralization to Reviewer 4), so including left-handers would add motor-system variance unrelated to the experimental manipulation. We now state this rationale in the Methods and cite supporting work:

In Methods, under "Participants" (page 29, lines 454-455):

*"We preregistered (<https://doi.org/10.17605/OSF.IO/K5P2M>) a target sample size of 35 participants, based on a small to medium effect size of $f \sim .35$, with an alpha level of .05 and a power of 80% (calculated using G*Power 3.1). **We restricted inclusion to right-handed adults to reduce variance from hemispheric dominance and lateralization**⁶⁵⁻⁷¹."*

6. The subjects practiced the saccade task for 15 minutes immediately prior to their MRI. Would this influence the recorded GABA+ levels?

We appreciate the opportunity to clarify the timing. We believe that, given the timing of the practice relative to MRI start, this is unlikely to have influenced GABA+ levels at time of MRS. It is important to note that there was a ~40-minute gap between the end of the brief practice and the start of the MRS acquisition (~10 min transition/prep + structural scans before MRS). Task- or learning-related GABA modulation, as measured with MRS, is generally transient and region-specific, occurring during performance/learning and beginning to recover within tens of minutes after cessation. For example, in the motor cortex, GABA+ decreases during motor learning and shows partial recovery within ~20 min post-task, consistent with short-lived changes rather than sustained shifts at rest (Chen et al., 2017; Floyer-Lea et al., 2006; Kolasinski et al., 2019). Moreover, several reviews/meta-analyses emphasize that post-task delays allow neurometabolites to return toward baseline, and that resting GABA+ measured with edited MRS exhibits good test-retest stability over much longer intervals (Li et al., 2024; Maes et al., 2022; Porges et al., 2021). Given our ~40-minute delay and that

MRS was acquired at rest, we believe it is unlikely that the brief practice affected the baseline GABA+ estimates. We have now also added this reasoning in the Methods section.

In Methods, under "MRS data acquisition" (page 36, lines 605-608):

"During MRS acquisition, participants were instructed to close their eyes. MRS data acquisition started approximately 40 minutes after the end of the task practice participants completed prior to their MRI scan, thus making it unlikely that the brief task practice biased our baseline GABA+ estimates⁸⁴⁻⁹⁰."

While we do not have evidence or prior suggestions that the MRS estimation was unduly influenced by the task practice preceding 40 minutes earlier, if one would expect an effect, this could potentially have led to a reduction in the inter-individual variability of MRS estimation, as all participants had the same experiential history preceding the MRS measurement. Furthermore, any potential bias would not have affected our critical analysis of the contrast between sham and TUS conditions.

7. Authors report I_{sppa} and MI. What about the free field pressure itself? I_{spta} ?

Thank you for pointing out this omission. We now report the free-field peak negative pressure and I_{spta} alongside I_{sppa} and MI. We have also added a full report based on "ITRUSST Consensus on Standardised Reporting for Transcranial Ultrasound Stimulation" in Figure S12.

In Methods, under "TUS protocol" (page 37, lines 631-636):

"The TUS protocol was adapted from Kubanek et al. (2020) (pulse duration: 2 ms; pulse ramp length: 1 ms; pulse repetition frequency: 500 Hz; pulse train duration: 500 ms; **square-wave equivalent** I_{sppa} in free water: 25 W/cm²; **effective I_{sppa} in free water: 9.37 W/cm²; I_{spta} in free water: 9.37 W/cm²; free-field pressure: 0.9 MPa; MI: 1.76; MI_{tc}: 1.25; Figure 1D, Figure S12).**

Transducers and drive systems description

	model number, manufacturer	centre frequency	radius of curvature	aperature diameter	number elements	element distribution
transducer	NeuroFUS CTX250-009, Sonic Concepts	250 kHz	63.2 mm	45.5 mm	2	Annular array comprising a circular innter element and an annular outer element of equal area
matching integrated drive system	NeuroFUS TPO-105, Sonic Concepts					
	model number, manufacturer	centre frequency	radius of curvature	aperature diameter	number elements	element distribution
transducer	NeuroFUS CTX250-014, Sonic Concepts	250 kHz	63.2 mm	45.5 mm	2	Annular array comprising a circular innter element and an annular outer element of equal area
matching integrated drive system	NeuroFUS TPO-203, Sonic Concepts					

Drive system settings

operating frequency	output level setting	focal positioning setting
250 kHz	25 W/cm ² (Isppa)	32.5 mm (FLHM)

Free field acoustic parameters

	model number, manufacturer	spatial-peak pressure amplitude	axial position spatial-peak pressure	position of centre of axial -3dB pressure	axial -3dB width	axial -6dB width
transducer	NeuroFUS CTX250-009, Sonic Concepts	0.9 kPa	30 mm	33 mm	25 mm	37 mm
transducer	NeuroFUS CTX250-014, Sonic Concepts	0.9 kPa	31 mm	34 mm	23 mm	34 mm

Pulse timing parameters

	duration	ramp duration	ramp shape	repetition interal / frequency
pulse	2 ms	1000 μs	tukey	2 ms / 500 Hz
pulse train	500 ms			

Intensity parameters

spatial-peak pulse-average intensity	spatial-peak time-average intensities	acoustic impedance
9.37W/cm ²	9.37 W/cm ²	1.5x10 ⁶ Rayls

Figure S12 | Transcranial ultrasonic stimulation standardized reporting details

8. The authors, in the discussion, claim “TUS effects are dependent on individual baseline GABA+ levels.” Is it appropriate to assert causality from the included data? Would it be more accurate to say they are predicted by or correlated with?

We agree that the suggested phrasing is more appropriate and have revised our phrasing throughout the manuscript:

In Results, §2.2 header (page 8, line 120):

*“FEF-TUS shows contralateral bias **correlated with** GABA+ levels”*

In Discussion, main text (page 19, line 262):

*“Importantly, the effect of FEF TUS **is associated with** individual inhibitory tone, as indexed with magnetic resonance spectroscopy (MRS).”*

In Discussion, header (page 22, line 331):

*“TUS effects are **correlated with** individual baseline GABA+ levels”*

In Discussion, main text (page 22, line 335):

*“The effects of TUS **can, therefore, be predicted by** the baseline state of the neuronal populations involved.”*

9. The authors performed their analysis with linear mixed effect models. These models have certain underlying assumptions about data structure (i.e., normality of residuals, homoscedasticity, linearity, etc). Were these assumptions checked?

Thank you for addressing this. This is all in order. GLMM assumptions were checked with DHARMA: residual uniformity (KS test), dispersion, and residual patterns vs. predictors for the main models. We have now added this procedure to the Methods section and its outcomes in Supplementary Documents S4 and Figure S10 (see below).

In Methods, under “Data Analysis” (page 39, lines 686-688):

“For binomial generalized linear mixed models we used simulation-based residual diagnostics (DHARMA) to verify model assumptions (Supplementary Documents S.4 & Figure S10).”

In Supplementary Documents S.4 (supplementary document, lines 114-130):

“For binomial generalized linear mixed models we used simulation-based residual diagnostics (DHARMA) to verify model assumptions. We assessed overall model fits by inspecting the uniformity of simulated residuals (DHARMA QQ/uniformity plot with a Kolmogorov-Smirnov test), and tested for (over/under)dispersion and outliers using DHARMA’s non-parametric procedures. To evaluate homoscedasticity and potential model mis-specification, we plotted residuals versus fitted values (rank-transformed). Linearity of continuous predictors (e.g., SOA_scaled) was examined both visually via residuals versus predictor plots and formally by comparing the linear specification with a natural-spline alternative (likelihood-ratio test). We also inspected residual

distributions across factor levels (e.g., stimulation side) using within-group uniformity checks and a Levene test for equality of variances. Finally, we evaluated multicollinearity of fixed effects via variance inflation factors (VIFs). All diagnostics were implemented in R with the packages DHARMA, performance/see, and influence.ME. Together, these diagnostics indicated adequate fit for both the left vs. right FEF model, and the Side × Region model. Residuals are approximately uniform with no over/under-dispersion or outlier excess, no systematic trend versus fitted values or SOA, and no heteroscedasticity. See Figure S10 for detailed results."

Figure S10 | GLMM diagnostics for two main statistical analyses

(A) *FEF-choice domain model*: DHARMA quantile–quantile (uniformity) plot of simulation-based residuals; Kolmogorov-Smirnov test: $p = 0.13389$ (n.s.); dispersion test: $p = 0.947$ (n.s.); outlier test: $p = 1$ (n.s.). (B) *FEF-choice domain model*: DHARMA residuals versus model predictions (both rank-transformed). The LOESS smooth (red) is approximately horizontal, indicating no global misfit or excess hetero(dis)persions beyond model expectation. (C) *FEF-choice domain model*: DHARMA residuals versus SOA_scaled (rank-transformed). The near-flat trend indicates no evidence of non-linearity in the logit with respect to SOA; a natural-spline alternative for SOA did not improve fit (LRT $\chi^2(2) = 0.52$, $p = 0.77$). (D) *FEF-choice domain model*: DHARMA residuals by stimSide (left FEF vs right FEF). Within-group deviation from uniformity: n.s.; Levene test for homogeneity of variance: n.s.; i.e., no heteroscedasticity across levels. (E) *Side × Region model*: DHARMA quantile–quantile (uniformity) plot of simulation-based residuals; Kolmogorov-Smirnov test: $p = 0.12922$ (n.s.); dispersion test: $p = 0.848$ (n.s.); outlier test: $p = 0.21016$ (n.s.). (F) *Side × Region model*: DHARMA residuals versus model predictions (both rank-transformed). The LOESS smooth (red) is approximately horizontal, indicating no global misfit or excess hetero(dis)persions beyond model expectation. (G) *Side × Region model*: DHARMA residuals versus SOA_scaled (rank-transformed). The near-flat trend indicates no evidence of non-linearity on the logit scale; a natural-spline alternative for SOA did not improve fit (LRT $\chi^2(2) = 0.626$, $p = 0.731$). (H) *Side × Region model*: DHARMA residuals by stimSide (left vs right). Within-group deviation from uniformity: n.s.; Levene test for homogeneity of variance: n.s. (I) *Side × Region model*: DHARMA residuals by stimRegion (FEF vs M1). Within-group deviation from uniformity: n.s.; Levene test for homogeneity of variance: n.s.

10. The linear mixed effect models used to analyze each metric are somewhat complex, often times with several fixed and random effects. Readers may benefit from a bit more discussion on what the equations mean in addition to the provided syntax.

Thank you for highlighting this. While we included the full syntax for transparency, we now realize that this may have unnecessarily complicated the presentation. We have therefore revised the section, as we now explicitly describe factor coding, z-scoring, and random effects structure up front, and then for all syntax described, leave out the random effects:

In Methods, under "Data analysis" (page 39, lines 677-682):

*"For all regression analyses reported below, SOA was included as a z-scored covariate, **and all factors were sum-to-zero coded**. To account for both between and within-subject variability, saccade data were analyzed with logistic mixed-effects models using the lme4 package in R69. **We included random effects for all within subject variables. Note that for simplicity, in the equations below, the random effect structure is not included in the syntax.**"*

Examples of a simplified syntax (line 726):

saccade direction ~ 1 + condition_{leftFEF/rightFEF} + SOA

*saccade direction ~ 1 + side_{left/right} * region_{FEF/M1} + SOA*

Other syntax lines: 705, 731, 737, 744, 746, 748, 770, 783, 795, 869, 871, 881, 883.

We hope that this addresses the reviewer's concerns.

11. For their primary analysis, it seems the authors made multiple models per each condition (for example: line 643 has a model specifically for left FEF vs. right FEF), and then analyzed each model separately for significance. However, in the supplementary analysis (line 689), they show a model including all FEF and M1 conditions together. The latter seems more intuitive for analysis, as it would directly allow for the emmeans pair-wise comparisons between all FEF, all M1, and all sham conditions. Why did they authors choose to not include all conditions in the same model(s) for their primary analysis?

Thank you for bringing this up. The reason that this is not part of our main analysis, is because our preregistered hypotheses target a contralateral bias specific to FEF, as these two FEFs are each other's most informative controls given their contralateral contributions to saccade selection. The most sensitive and interpretable confirmatory tests are therefore (i) left-vs-right FEF (line 726) and (ii) the side-by-region (FEF vs M1) interaction (line 737) as

an extra control analysis, which incorporates the full design while using M1 as an active control. However, we did fit an all-conditions model (including left/right FEF, left/right M1 and sham) in addition to the primary models, and reported the corresponding estimates and pairwise comparisons in Supplementary Tables S1-S9. We acknowledge that this was not clearly signposted in the Methods and have now made this explicit.

In Methods, under "TUS effects" (page 40, lines 712-714):

*"Choice domains were defined at the individual level by determining the delay windows, i.e. SOAs, where participants showed a probability of making rightward saccades between 0.25 and 0.75 (see Supplementary Tables S.1-S.9 for (i) **condition (all) and SOA, (ii) stimulation side (left/right) by stimulation region (FEF/M1) and SOA, and (iii) MRS effect analyses for this and other choice domain window results.**"*

12. The authors made many statistical comparisons, but did not seem to apply any kind of p-value multiple comparison correction. Given that multiple comparison correction seems to be the TUS field standard, they may want to elaborate on why they chose not to do so.

We agree that multiplicity should be handled transparently. In our case, the confirmatory analyses were defined a priori and preregistered: the primary family of analyses targets the contralateral-bias hypothesis via the left-vs-right FEF comparison, and as an additional control analysis the Side × Region (FEF vs M1) interaction. Because the number and nature of these tests were fixed before inspecting the data, a global multiple-comparison correction is not required; type-I error is controlled by design. Moreover, these two tests are not independent as they are estimated on the same dataset and probe overlapping hypotheses, so applying broad corrections as if tests were independent would be over-conservative. All further analyses (M1 left vs M1 right, auditory masking, TUS after-effects) are included as control analyses. Since the underlying hypothesis assumes that there are no effects, applying multiple comparisons corrections would unnecessarily reduce the sensitivity to detect problematic effects.

13. The authors present right M1 vs. left M1 behavioral comparisons in Figure 4, separately from the right FEF vs. left FEF. It may help strengthen their results if they can combine these results into a single figure to demonstrate that FEF stimulation was not only different from each other, but also different from the results of their spatial control.

We appreciate the suggestion. Our primary inference targets a side bias mechanism and is therefore based on the left-vs-right FEF contrast. In this framework, left and right FEF are each other's most informative controls because both contribute to contralateral saccade selection. M1 (left/right) serves as an active spatial control, which we test via the Side × Region (FEF vs M1) interaction. Sham functions as a passive control.

In line with our pre-registration, we suggest keeping Figure 2D in the main text organized around the FEF left-vs-right comparison. That said, we agree a direct visual comparison across conditions might be informative; therefore, we have added a combined panel with

all four TUS conditions (left/right FEF and left/right M1; common axes/binning), and a panel with left/right FEF and sham (as suggested by Reviewer 3) as Supplementary Figure S1:

Figure S1 | Empirical psychometric curves of FEF and control conditions

Left panel shows FEF vs sham TUS effects, right panel shows FEF vs M1 TUS effects. Data are binned for visualization purposes into bins of approximately equal numbers of trials, resulting in SOA intervals of [0], [8.3 - 25.0], [33.3 - 50.0], [58.3 - 75.0], [83.3 - 100.0], [108.3 - 141.7], and [150 - 200]. Bins are symmetric for negative values. Note that all inferential statistics were performed using continuous SOAs. Dots represent the group mean per bin, and error bars indicate the S.E.M. across participants.

14. One of the main claims in the Summary is that their work “highlight[s] the importance of neurophysiological state in neuromodulation.” However, in their discussion, they write, “TUS acts to normalize interindividual differences in physiological states.” Are these two statements contradictory? If TUS is normalizing the interindividual effects, wouldn’t the baseline neurophysiological state be unimportant?

Thank you for pointing out this potential ambiguity, and we agree our wording in the discussion was overly strong. We did not want to imply that baseline neurophysiological state is unimportant or that TUS “eliminates” individual differences. Instead, our key point is that our data suggest that TUS effects are baseline dependent: baseline FEF GABA+ predicts the magnitude of the TUS-induced bias (Results §2.2). Following TUS to FEF, saccade choices shift in a contralateral direction, but the degree to which this happens depends on baseline GABA+ levels. Participants with higher cortical inhibitory tone (baseline GABA+), which in our data predicted a larger contralateral bias under sham, showed a smaller TUS-induced shift; conversely, participants with lower inhibitory tone showed larger shifts. To avoid confusion, we have revised the discussion:

In Discussion (page 23, lines 355-358):

“Furthermore, baseline **physiological inhibitory state** predicted the magnitude of the TUS-induced behavioral effects, *such that participants with higher cortical inhibitory tone show a larger contralateral bias under sham, and exhibit a smaller TUS-induced shift, whereas those with lower inhibitory tone exhibit a larger shift* (Figure 3A). **These findings** underscore the state-dependent nature of TUS and highlight the importance of considering baseline neural states when interpreting its effects.”

15. The authors only had time to collect one hemisphere’s worth of GABA+ readings, so they are unable to see if baseline right FEF GABA+ is correlated with right TUS effect sizes. However, as an additional control, could they investigate whether left FEF GABA+ is correlated with right TUS effects?

This is indeed a potentially informative control for hemispheric specificity of the GABA+ effects. However, given evidence that GABA+ levels for homologous cross-hemisphere sensorimotor areas are strongly correlated (Puts et al., 2018), we a priori predicted that correlations may be similar in direction. This informed our decision that unilateral GABA+ estimates might be sufficient to test our hypotheses. Indeed, correlating left-hemisphere FEF GABA+ with the behavioral effect under right-FEF TUS resulted in a trend association (FEF_{right} vs Sham \times GABA+_{FEF}: $b = 0.10$, 95% CI $[-0.01, 0.20]$, $\chi^2 = 3.4$, $p = 0.07$). This aligns with prior evidence that interhemispheric GABA+ coupling is moderate and region-dependent and with lateralization in attention networks, including FEF (Grewal et al., 2018; Violante et al., 2016).

Reviewer #2 (Remarks to the Author):

We thank the reviewer for co-reviewing this manuscript. We appreciate their contribution and have addressed the points raised by their co-reviewer in detail.

Reviewer #3 (Remarks to the Author):

The authors develop the online TUS technique that allows trial-based comparisons within blocks. Although not highlighted sufficiently, in my opinion, it is quite interesting that various TUS after-effects can be elicited by auditory/somatosensory confounds depending on the experimental settings, which emphasizes the importance of appropriate control of experimental procedures. I would raise several points to be addressed that I hope will improve this manuscript.

We thank the reviewer for highlighting this important point. We fully agree that auditory/somatosensory co-stimulation can shape apparent after-effects and that appropriate control conditions should not be limited to a generic 'sham'. Following this suggestion, we now state this more explicitly in the Discussion and cite the relevant literature (including Duecker & Sack, 2015 who have already described this in the domain of TMS, and Kop et al., 2024 who emphasizes this need for TUS studies as well).

In Discussion (page 21, lines 300-303):

“Given that on a group-level, participants were able to distinguish TUS from sham trials, we highlight here that control conditions must go beyond a generic ‘sham’ and explicitly match peripheral confounds and expectancy; a consideration that is often overlooked in non-invasive brain stimulation^{9,48}.”

1. The stimulation pulse was structured, with fundamental frequency of 250 kHz, ramping up and down sinusoidally within 2 msec (up: 1 msec, down: 1 msec), and lasting for 500 msec, if I understand correctly. I think the excitatory effect of online TUS is rare and wondered which component of the structured stimulation pulse elicited excitatory effect on the FEF. It would also be interesting to see historical summary of how the stimulation protocol was developed.

We appreciate this thoughtful comment and the opportunity to clarify our interpretation and protocol rationale. We avoid characterizing TUS as intrinsically “excitatory” or “inhibitory”. Net effects are parameter-, physiology-, and state-dependent and reflect the local circuitry engaged at the time of stimulation: for example, a net excitatory effect could arise from either excitation of glutamatergic neurons or inhibition of GABAergic neurons. The current literature does not support a one-size-fits-all “excitatory” recipe independent of the target region and brain state. This is now made explicit in the discussion.

In Discussion, (page 20, lines 289-296):

“Taken together, the directionality and context of the effect are consistent with a net facilitatory effect on FEF-mediated contralateral selection, replicating earlier work in non-human primates². Note that we deliberately avoid characterizing TUS as intrinsically “excitatory” or “inhibitory”. Net effects are parameter-, physiology-, and state-dependent and reflect the local circuitry engaged at the time of stimulation: for example, a net excitatory effect could arise from either excitation of glutamatergic neurons or inhibition of GABAergic neurons. The current literature does not support a one-size-fits-all “excitatory” recipe independent of the target region and brain state.”

Our protocol builds on prior non-human primate work (Kubanek et al., 2020), which used a closely related pulse family and reported online, facilitatory shifts in choice behavior during stimulation. The core rationale was already present in our Methods (lines 640 - 650), but to

make this explicit, we have added some clarifying sentences that state our human-specific adaptations (ramped bursts with auditory masking, a longer pulse train, and higher free field intensity to offset skull attenuation).

In Methods, under “TUS protocol” (page 37, lines 631-650):

*“The TUS protocol was adapted from Kubanek et al. (2020) (pulse duration: 2 ms; pulse ramp length: 1 ms; pulse repetition frequency: 500 Hz; pulse train duration: 500 ms; **square-wave equivalent I_{sppa} in free water: 25 W/cm²; effective I_{sppa} in free water: 9.37 W/cm²; I_{spta} in free water: 9.37 W/cm²; free-field pressure: 0.9 MPa; MI: 1.76; MI_{tc}: 1.25; Figure 1D, Figure S12).***

Relative to the original simulation protocol in non-human primates, we used ramped pulses with an auditory mask to minimize auditory co-stimulation, increased free-water I_{sppa} to offset skull attenuation, and a longer pulse train duration to match human saccade preparation and execution.

Although squared and sinusoidal ramped pulses have the same integral energy content, it is important to note that squared wave pulses have associated limitations. A squared pulse encompasses a constant intensity peak for a longer duration due to their clear onset and offset, whereas a sinus-shaped pulse exhibits a gradually increasing and decreasing peak that is never fully off. While low-intensity ultrasonic waves are beyond the range of human hearing, the on-offset of the squared pulse is detectable by humans, increasing the likelihood of auditory confounds, and thus contributing to a clearer temporal profile of stimulation^{4,79}. Furthermore, since humans have a thicker skull than macaques, a higher free-field I_{sppa} was applied (25 W/cm²) to match the realized intracranial intensity across species. Moreover, we adjusted the total stimulation duration to the average human saccade duration.”

2. If it is possible to estimate, based on the eye movement results, how robustly the stimulation protocol can modulate behavior in cognitive tasks, that would be helpful.

On a neuronal level one could expect the induced effects to be comparable across studies when sonicating regions neurophysiologically and morphologically similar to the cortical region we targeted in this study. However, the behavioral robustness observed in our eye-movement paradigm might not translate directly to other procedures with different behavioral measures. To facilitate future comparisons, we now report effect sizes—specifically, within-subject standardized effect sizes (with 95% CIs) for our two primary analyses: the left-vs-right FEF contrast and the Side × Region (FEF vs M1) interaction.

In Results, §2.2 (page 8, lines 122-126):

“Left vs right FEF increased the odds of a rightward saccade by 13% (OR=1.13, 95% CI 1.05-1.22), corresponding to Cohen’s d (logit) = 0.07 (95% CI 0.03-0.11). At the participant level, the within-subject SMD of model-predicted probabilities was dz = 1.27 (Hedges’ g = 1.24; n=35), indicating a consistent modulation across subjects.”

In Results, §2.3 (page 12, lines 173-175):

“The side-by-region model demonstrated stronger side modulation in FEF than in M1 by 26% (ROR = 1.26, 95% CI 1.03-1.56), corresponding to Cohen’s d (logit) = 0.13 (95 CI 0.01-0.24) indicating a small but reliable interaction.”

3. It would be informative to show raw data of eye movements and how saccade direction was determined.

We have now added more information on raw data and saccade processing. We specify in the methods how the first qualifying saccade was determined and how direction (left vs. right) was defined from the endpoint ROI. We also included visualizations of example trial traces (with the selected saccade indicated) and 2D heatmaps of start positions and endpoints illustrating the fixation window and left/right ROIs in Supplementary Figure S9.

In Methods, under “Behavioral acquisition” (page 38, lines 659-670):

“Saccades were detected by the eye tracker’s online velocity/acceleration parser using the standard cognitive-task thresholds (velocity 30°/s, acceleration 8000°/s², motion criterion of 0.15°). From the exported raw data file, we used the Python-generated event markers to identify trial boundaries and target onset, and then assigned each saccade to its corresponding trial. Note that this could result in multiple saccades per trial. Only the first qualifying saccade per trial was included in the statistical analyses. The first qualifying saccade after the target was defined by meeting the following criteria: (i) the start position within a central fixation window ($x \in [860, 1060]$ px), (ii) the endpoint within one of two lateral ROIs (Left ROI: $x \in [0, 800]$ px; Right ROI: $x \in [1120, 1920]$ px), and (iii) no blink overlapped the saccade. Direction was defined by the endpoint ROI—landings in the right ROI were labeled right and in the left ROI left. The selection criteria and raw saccadic data are illustrated in Figure S9.”

Figure S9 | Raw saccade examples and spatial distributions (example participant)

(A) Heatmaps of saccade start positions (left) and endpoints (right) across all qualifying trials. Bright colors indicate higher density estimates. Blue overlays show the task visual targets: fixation at screen center (960 px) and left/right targets at ±10° from center (crosses). (B) Velocity-time traces (°/s) for the same example trials, aligned to target onset ($t = 0$ ms). Shaded rectangles (matching the line colors) indicate the detected saccade duration for each trial; peaks correspond to primary and, when present, corrective saccades. The horizontal dashed line denotes the velocity threshold used for saccade detection. (C) Overlaid raw examples of the first qualifying saccade after target onset from several trials. Filled circles mark saccade onset; × marks the endpoint. Thin lines show the raw gaze trace within each trial. Colors encode saccade direction (blue = leftward; red = rightward). Axes are gaze position in pixels.

4. Three hypothetical TUS effects (excitatory, inhibitory, and perturbatory shown in Fig. 1E) have distinct patterns, but unfortunately the curve for the sham condition was not shown in Fig. 2D.

We appreciate the suggestion. Our primary inference targets a contralateral (side) bias mechanism and is therefore based on the left-vs-right FEF contrast. In this framework, left and right FEF are each other's most informative controls because both contribute to contralateral saccade selection. Considering our pre-registered analyses, we suggest keeping Figure 2D in the main text organized around the FEF left-vs-right comparison. However, we appreciate the reviewers' point and did include i) the statistical comparison details of left and right FEF relative to sham in Figure 2C, and ii) present the raw sigmoid

functions for both FEF and conditions in Supplementary Figure S1. We also added the curves overlaying both FEF and M1 conditions as suggested by Reviewer 1.

Figure S1 | Empirical psychometric curves of FEF and control conditions
Left panel shows FEF vs sham TUS effects, right panel shows FEF vs M1 TUS effects. Data are binned for visualization purposes into bins of approximately equal numbers of trials, resulting in SOA intervals of [0, [8.3 - 25.0], [33.3 - 50.0], [58.3 - 75.0], [83.3 - 100.0], [108.3 - 141.7], and [150 - 200]. Bins are symmetric for negative values. Note that all inferential statistics were performed using continuous SOAs. Dots represent the group mean per bin, and error bars indicate the S.E.M. across participants.

In Figure 2C (page 10):

(C) FEF TUS effects. Choice-domain average effects. Grey dots represent individual participants' mean saccadic directions within the choice domain. Colored dots represent group means, error bars indicate the standard error of the mean (S.E.M.). *** $p < 0.001$, * $p < 0.05$, † $p < 0.1$.

5. Was the mysterious biased saccade direction around zero SOA after M1 stimulation observed in previous studies of this kind?

Like the reviewer, we were puzzled by this zero-SOA M1 effect. To our knowledge, prior human brain-stimulation work—particularly M1 TMS in visual/perceptual paradigms—typically reports no lateralized effects on saccade choice and often no behavioral impact comparable to ours (Coubard & Kapoula, 2006; Kapoula et al., 2004; Mathew et al., 2017). However, those studies generally employed single-target or pursuit tasks and focused on latency/kinematics, rather than two-alternative, near-threshold decisions. One could speculate that the 0-delay trials are perceived as qualitatively different even from delay trials (even when very short). Humans are highly sensitive to detecting ‘flicker’, and thus, instantaneous presentations of the two targets may have been easy to detect (Foerster et al., 2025; Mankowska et al., 2021). If participants would (correctly) identify this condition as ‘simultaneous’, they could realize that there was no ‘correct’ response in this condition and thus trigger some attentional response. That said, it is important to note that the FEF TUS effects remain constant and stable also in the SOA = 0 bin, and only the M1 0-SOA bin is an outlier.

We now add the following sentence to the **Results, §2.3 (page 12, lines 165-167)**:

“We remark that these zero-delay trials are qualitatively different from the delay trials because of simultaneous target presentation, which may be easily identified and thus trigger different cognitive processing^{37,38}.”

6. The saccade direction in sham trials following sham trials appeared biased toward the left side (Fig6A). If the left side bias is a general natural tendency, an alternative interpretation for the after-effect assessment results would be possible.

The TUS after-effect analysis contains a limited number of trials (mean = 22 trials per condition, SD = 7, range 6-43; Methods, lines 788-789), so any results here should be treated with caution, as these analyses are exploratory and underpowered. Nevertheless, to address the reviewer’s point, we tested the sham→sham condition, and found no significant bias. We have now added this to **Supplementary Documents S3**:

*“To investigate potential longer-lasting TUS effects beyond the stimulation duration itself, we analyzed sham trials that directly followed a TUS trial. **First, as a baseline, we tested the sham→sham condition and found no significant bias following sham trials, with a non-significant intercept on the logit scale ($b = -0.07$, $p = 0.3$), corresponding to $p = 0.5$ (95%-CI [0.455-0.512]) from 0.5.** Interestingly, **for sham trials following TUS**, we observed a significant increase in ipsilateral responses for example, if a sham trial followed a left TUS trial, participants were more likely to make a leftward saccade ($side_{t-1}$: $b = -0.10$, 95%-CI [-0.18, -0.02], $\chi^2 = 5.3$, $p = 0.021$; Figure 6A).”*

We would like to emphasize that the same small side tendency is identical following both FEF and M1 TUS trials, which argues for a non-specific attentional/arousal influence, potentially induced by the sound elicited by TUS on the preceding trial, rather than by neural effects directly induced by ultrasound stimulation (further discussed in Supplementary Documents S3). Participants could detect stimulation above chance but reported the side of stimulation to be contralateral to the actual stimulation side (Figure 6C). As for the after-effects, unlike the key findings, these effects are independent of the stimulation site (FEF or M1). As such, these effects do not confound our primary findings.

Results, §2.4 (page 17, lines 249-250):

*"Taken together, while putative confounding factors were observed in our study, **their effects were present in both FEF and M1 conditions, and thus crucially cannot account for our main findings. More broadly, this masking assessment emphasizes the importance of active control conditions for online TUS protocols.**"*

Reviewer #4 (Remarks to the Author):

Farboud et al describe experiments in which they test the hypothesis that (relatively) short (500 ms) transcranial ultrasound (TUS) pulses delivered to the frontal eye field (FEF) in humans should bias saccadic eye movements as they appear to in nonhuman primates (e.g. ref. 2). The authors point out that the large majority of human studies to date focus on repetitive 'offline' protocols with temporally sustained effects of TUS, while those in animals aim for short effects, which provide great utility in addressing more circuit-level effects. Using the short protocol, the authors indeed report that TUS delivered the FEF (left and right) increase the probability of contralateral saccades, like results in nonhuman primates. In addition, the authors measured GABA levels in the (left) FEF via magnetic resonance spectroscopy (MRS) and found that the level of GABA predict the bias in saccades as well as the effects of TUS. Overall, the results are relatively straightforward and fairly convincing. I only have a few comments/suggestions that I believe would improve the interpretation and soundness of the results.

Major comments.

1. The authors aim to emulate a previous study of ultrasound in nonhuman primates (ref. 2), yet the task employed differs in an important way. In the previous study (and others), subjects perform a free-choice task in which they can choose either target. However, the onset asynchrony typically causes them to choose the first target. Nonetheless, subjects (monkeys or humans) are not instructed to saccade to the first target, as in this study. While I do not think this is a critical issue, I do think it exposes an important point regarding the interpretation of the present results. In the free-choice task, one might argue that biases in saccades elicited by stimulation are more related to motor control and motor decisions. In contrast, when the task is explicitly a perceptual one, i.e. 'saccade to the one perceived as

appearing first' one might argue that the bias effects are more likely to reflect changes in perception.

The distinction, while not crucial here, is important given that the FEF has an established role in the control of visuospatial attention in both human and nonhuman primates (e.g. <https://www.pnas.org/doi/abs/10.1073/pnas.98.3.1273>; <https://ieeexplore.ieee.org/abstract/document/6789738>) and has been shown to control sensory representations in visual cortex (e.g. <https://www.nature.com/articles/nature01341>; [https://www.cell.com/current-biology/fulltext/S0960-9822\(06\)01818-5?cc=y%3Fcc=y](https://www.cell.com/current-biology/fulltext/S0960-9822(06)01818-5?cc=y%3Fcc=y)). Thus, an alternative interpretation of the effects of TUS on FEF observed would be that subjects perceived the onset in the contralateral hemifield as earlier than without TUS due to attentional deployment. Since it has been shown that attention affects temporal order judgments (e.g. <https://psycnet.apa.org/fulltext/1991-26439-001.html>), this interpretation is entirely sound. Moreover, the effects would be expected to be more robust within the so-called 'choice-domain' that the authors focus on (Figure 2D). Again, I don't think this bears critically on the importance of the results described, but they authors definitely need to modify/expand their conclusions to more accurately reflect what is known about the FEF's role in visually guided behavior.

We thank the reviewer for this thoughtful, incisive point. It provides indeed an alternative interpretation of our findings, and we have revised the Discussion accordingly to incorporate this perspective (using the reviewers suggested references and very eloquent comments).

In Discussion (pages 19 - 20, lines 270-291; page 21, lines 305-320):

"In recent years, evidence has emerged for sustained and early-phase plasticity effects of TUS in humans^{7,8,43}. Despite advances in in-vitro and animal models demonstrating immediate neural effects of TUS^{2,4,6}, the translation of these findings to humans has remained scarce¹⁰ or potentially marred by confounds⁹. To address this, we adapted a well-established animal TUS protocol for human application, targeting the left and right FEF while participants performed a saccade choice task. This approach allowed us to assess the immediate behavioral effects of TUS on saccadic choices.

*Previous lesion and brain stimulation studies have demonstrated the **key role of FEF in oculomotor control**, mediating contralateral saccade generation¹³⁻²⁴. Here, we reveal that TUS over FEFs increased the selection of contralateral saccades, particularly during **with short SOAs** (Figure 2C-D). **These findings, in conjunction with this past literature, suggest that the saccade bias elicited by TUS reflects a modulation of the oculomotor control by the FEF. An alternative interpretation of the results, inspired by a discussion with a reviewer, may be that the bias effects reflect changes in perception itself. The FEF has an established role in the control of visuospatial attention^{17,44} and control of sensory representations in visual cortex^{45,46}. Considering that attention affects***

temporal order judgments⁴⁷, a complementary interpretation would be that FEF TUS altered the perceived temporal priority. Regardless, our results emphasize TUS induced changes in the sensorimotor transformations performed by the FEF in the context of eye movements.

Taken together, the directionality and context of the effect are consistent with a net facilitatory effect on FEF-mediated contralateral selection, replicating earlier work in non-human primates².

...

Our results show that TUS biases responses on trials with short SOAs, where there was high uncertainty about the correct response (Figure 2D). On those trials, which we purposefully oversampled, we hypothesized FEF activity is more sensitive to be 'nudged' by TUS to tip the balance in the opposite direction. This effect supports the hypothesized role of TUS in modulation of ongoing FEF computations, rather than the direct induction of saccades. TUS employs sound waves that are beyond the audible spectrum to mechanically engage with neuronal tissue, influencing characteristics such as membrane capacitance and the activity of mechanosensitive ion channels⁶. Mechanistically, unlike magnetic or electrical stimulation techniques that can directly induce neural firing (motor evoked potentials³³; visual evoked potentials^{34,35}), TUS does not directly evoke neural firing but rather modulates ongoing neural activity through subthreshold modulation and subtly 'nudges' neural activity without inducing immediate or large-scale neural responses³⁶. **Thus, two principles converge here: low-evidence trials are most susceptible to small motor biases, and temporal-order judgments are sensitive to perceptual and attentional biases^{17,44-47}. Together, these predict that TUS effects should peak at short SOAs. Consistent with this, we observe the largest biases in this short SOA regime."**

2. The differences in magnitude and variability of TUS effects on saccades shown in figure 2C appear to be noteworthy. Although the authors do briefly refer to differences in lateralization of function between humans and animals (discussion) they don't appear to make much of the apparent differences between left and right FEF TUS (e.g. <https://www.sciencedirect.com/science/article/pii/S2352250X18302392>). Perhaps it would be worth addresses this in their interpretation of the results and the extent to which their results are consistent with past evidence.

This is a sharp observation and suggestion. We had noticed this too, but were initially reluctant to over-interpret our results, as our analyses were pre-registered to be focused on the left- vs. right-FEF contrast. However, inspired by the reviewer, we have now slightly expanded on this observation in the Discussion.

In Discussion (pages 23-24, lines 365-373):

*“Notably, the effects of TUS observed in humans are less pronounced than those documented in earlier animal work². **At least two reasons** can be conceived. First, human FEF is larger in absolute terms, reducing the relative volume of FEF reached by the TUS focus. This might lead to a reduced efficacy of stimulation, as the average TUS intensity across the entire FEF will be lower in humans than in macaques, even with comparable peak intensities. **Second, compared the FEF circuitry in animals is more lateralized compared to humans^{58,59}, which may lead to weaker effects of particularly right FEF stimulation. A left-right asymmetry is plausible given the well-documented right-hemisphere dominance of human attention networks and hemispheric differences in fronto-parietal circuitry⁶⁰⁻⁶³.”***

- Anderson, E. J., Jones, D. K., O’Gorman, R. L., Leemans, A., Catani, M., & Husain, M. (2012). Cortical Network for Gaze Control in Humans Revealed Using Multimodal MRI. *Cerebral Cortex (New York, NY)*, 22(4), 765-775. <https://doi.org/10.1093/cercor/bhr110>
- Butler, C. R., Rhodes, E., Blackmore, J., Cheng, X., Peach, R. L., Veldsman, M., Sheerin, F., & Cleveland, R. O. (2022). Transcranial ultrasound stimulation to human middle temporal complex improves visual motion detection and modulates electrophysiological responses. *Brain Stimulation*, 15(5), 1236-1245. <https://doi.org/10.1016/j.brs.2022.08.022>
- Chen, C., Sigurdsson, H. P., Pépés, S. E., Auer, D. P., Morris, P. G., Morgan, P. S., Gowland, P. A., & Jackson, S. R. (2017). Activation induced changes in GABA: Functional MRS at 7 T with MEGA-sLASER. *NeuroImage*, 156, 207-213. <https://doi.org/10.1016/j.neuroimage.2017.05.044>
- Coubard, O. A., & Kapoula, Z. (2006). Dorsolateral prefrontal cortex prevents short-latency saccade and vergence: A TMS study. *Cerebral Cortex (New York, N.Y.: 1991)*, 16(3), 425-436. <https://doi.org/10.1093/cercor/bhi122>
- Duecker, F., & Sack, A. T. (2015). Rethinking the role of sham TMS. *Frontiers in Psychology*, 6, 210. <https://doi.org/10.3389/fpsyg.2015.00210>
- Floyer-Lea, A., Wylezinska, M., Kincses, T., & Matthews, P. M. (2006). Rapid modulation of GABA concentration in human sensorimotor cortex during motor learning. *Journal of Neurophysiology*, 95(3), 1639-1644. <https://doi.org/10.1152/jn.00346.2005>
- Foerster, F. R., Giersch, A., & Cleeremans, A. (2025). Spatial but not temporal orienting of attention enhances the temporal acuity of human peripheral vision. *Communications Psychology*, 3(1), 116. <https://doi.org/10.1038/s44271-025-00295-6>

Grewal, S. S., Middlebrooks, E. H., Kaufmann, T. J., Stead, M., Lundstrom, B. N., Worrell, G. A., Lin, C., Baydin, S., & Van Gompel, J. J. (2018). Fast gray matter acquisition T1 inversion recovery MRI to delineate the mammillothalamic tract for preoperative direct targeting of the anterior nucleus of the thalamus for deep brain stimulation in epilepsy. *Neurosurgical Focus*, *45*(2), E6. <https://doi.org/10.3171/2018.4.FOCUS18147>

Johnstone, L. T., Karlsson, E. M., & Carey, D. P. (2021). Left-Handers Are Less Lateralized Than Right-Handers for Both Left and Right Hemispheric Functions. *Cerebral Cortex*, *31*(8), 3780-3787. <https://doi.org/10.1093/cercor/bhab048>

Kapoula, Z., Yang, Q., Coubard, O., Daunys, G., & Orssaud, C. (2004). Transcranial magnetic stimulation of the posterior parietal cortex delays the latency of both isolated and combined vergence-saccade movements in humans. *Neuroscience Letters*, *360*(1), 95-99. <https://doi.org/10.1016/j.neulet.2004.01.077>

Kolasinski, J., Hinson, E. L., Divanbeighi Zand, A. P., Rizov, A., Emir, U. E., & Stagg, C. J. (2019). The dynamics of cortical GABA in human motor learning. *The Journal of Physiology*, *597*(1), 271-282. <https://doi.org/10.1113/JP276626>

Kop, B. R., Oghli, Y. S., Grippe, T. C., Nandi, T., Lefkes, J., Meijer, S. W., Farboud, S., Engels, M., Hamani, M., Null, M., Radetz, A., Hassan, U., Darmani, G., Chetverikov, A., Ouden, H. E. M., den, Bergmann, T. O., Chen, R., & Verhagen, L. (2024). Auditory confounds can drive online effects of transcranial ultrasonic stimulation in humans. *eLife*, *12*. <https://doi.org/10.7554/eLife.88762.2>

Kubanek, J., Brown, J., Ye, P., Pauly, K. B., Moore, T., & Newsome, W. (2020). Remote, brain region-specific control of choice behavior with ultrasonic waves. *Science Advances*, *6*(21), eaaz4193. <https://doi.org/10.1126/sciadv.aaz4193>

- Li, H., Rodríguez-Nieto, G., Chalavi, S., Seer, C., Mikkelsen, M., Edden, R. A. E., & Swinnen, S. P. (2024). MRS-assessed brain GABA modulation in response to task performance and learning. *Behavioral and Brain Functions: BBF*, 20, 22. <https://doi.org/10.1186/s12993-024-00248-9>
- Maes, C., Cuypers, K., Peeters, R., Sunaert, S., Edden, R. A. E., Gooijers, J., & Swinnen, S. P. (2022). Task-Related Modulation of Sensorimotor GABA+ Levels in Association with Brain Activity and Motor Performance: A Multimodal MRS-fMRI Study in Young and Older Adults. *Journal of Neuroscience*, 42(6), 1119-1130. <https://doi.org/10.1523/JNEUROSCI.1154-21.2021>
- Mankowska, N. D., Marcinkowska, A. B., Waskow, M., Sharma, R. I., Kot, J., & Winklewski, P. J. (2021). Critical Flicker Fusion Frequency: A Narrative Review. *Medicina*, 57(10), 1096. <https://doi.org/10.3390/medicina57101096>
- Mathew, J., Eusebio, A., & Danion, F. (2017). Limited Contribution of Primary Motor Cortex in Eye-Hand Coordination: A TMS Study. *The Journal of Neuroscience*, 37(40), 9730-9740. <https://doi.org/10.1523/JNEUROSCI.0564-17.2017>
- Pool, E.-M., Rehme, A. K., Fink, G. R., Eickhoff, S. B., & Grefkes, C. (2014). Handedness and effective connectivity of the motor system. *NeuroImage*, 99, 451-460. <https://doi.org/10.1016/j.neuroimage.2014.05.048>
- Porges, E. C., Jensen, G., Foster, B., Edden, R. A., & Puts, N. A. (2021). The trajectory of cortical GABA across the lifespan, an individual participant data meta-analysis of edited MRS studies. *eLife*, 10, e62575. <https://doi.org/10.7554/eLife.62575>

Puts, N. A. J., Heba, S., Harris, A. D., Evans, C. J., McGonigle, D. J., Tegenthoff, M., Schmidt-Wilcke, T., & Edden, R. A. E. (2018). GABA Levels in Left and Right Sensorimotor Cortex Correlate across Individuals. *Biomedicines*, *6*(3), 80. <https://doi.org/10.3390/biomedicines6030080>

Shulman, G. L., Pope, D. L. W., Astafiev, S. V., McAvoy, M. P., Snyder, A. Z., & Corbetta, M. (2010). Right Hemisphere Dominance during Spatial Selective Attention and Target Detection Occurs Outside the Dorsal Frontoparietal Network. *The Journal of Neuroscience*, *30*(10), 3640-3651. <https://doi.org/10.1523/JNEUROSCI.4085-09.2010>

Tomasi, D., & Volkow, N. D. (2024). Associations between handedness and brain functional connectivity patterns in children. *Nature Communications*, *15*(1), 2355. <https://doi.org/10.1038/s41467-024-46690-1>

Violante, I. R., Patricio, M., Bernardino, I., Rebola, J., Abrunhosa, A. J., Ferreira, N., & Castelo-Branco, M. (2016). GABA deficiency in NF1: A multimodal [¹¹C]-flumazenil and spectroscopy study. *Neurology*, *87*(9), 897-904. <https://doi.org/10.1212/WNL.0000000000003044>

Vossel, S., Geng, J. J., & Fink, G. R. (2014). Dorsal and Ventral Attention Systems. *The Neuroscientist*, *20*(2), 150-159. <https://doi.org/10.1177/1073858413494269>

Yoo, S., Mittelstein, D. R., Hurt, R. C., Lacroix, J., & Shapiro, M. G. (2022). Focused ultrasound excites cortical neurons via mechanosensitive calcium accumulation and ion channel amplification. *Nature Communications*, *13*(1), 493. <https://doi.org/10.1038/s41467-022-28040-1>

Rapid modulation of choice behavior by ultrasound on the human frontal eye fields

Soha Farboud, Benjamin R. Kop, Renée S. Koolschijn, Solenn L.Y. Walstra, José P. Marques, Andrey Chetverikov, W. Pieter Medendorp, Lennart Verhagen, Hanneke E.M. den Ouden

REVIEWER COMMENTS

We thank the reviewers for their efforts, and reviewer 1 for their final thoughtful comments, which we address below. In our reply, we use the following font styles for clarity:

- reviewer text in blue,
- reply to reviewers in normal font,
- *"manuscript text in italics and orange"*
- ***"changes in the manuscript text in bold, italics and orange"***

REVIEWERS' COMMENTS

Reviewer #1 (Remarks to the Author):

Farboud et al have addressed the raised questions thoughtfully. Their manuscript is well written, and their figures are informative. Their results are straightforward, and consistent with prior work done in nonhuman primates.

My remaining concern is with the strength of the data relative to the control condition(s). The authors have now provided a direct pairwise comparison between the audio mask sham condition and the FEF-targeted tFUS conditions. However, these direct comparisons are not particularly favorable ($p < 0.1$ for right FEF vs. sham).

Similarly, for the now-provided comparison on FEF and M1 targeted tFUS model, the Cohen's d effect size is only 0.13. I think the usual interpretations for Cohen's d values consider weak effects ≥ 0.2 . Less than 0.2 is, to my knowledge, very tiny or even negligible.

Given that the direct comparisons between the FEF targeted tFUS and the control conditions are not all that strong, perhaps the authors should tone down their language throughout the manuscript (they refer to their findings as "robust" numerous times).

We thank the reviewer for raising these points and for the opportunity to clarify our interpretation.

First, in response to the reviewer's request, we have toned down the strength of our language, at various points replacing "robust" with more neutral wording where appropriate:

Results, lines 134-136:

*"These results highlight the specificity of the effects to the FEFs and provide **robust***

converging evidence of direct TUS-induced behavioral changes in humans.”

Discussion, header line 266:

“FEF-specific TUS effects are fast and induce **robust** contralateral bias”

Discussion, lines 390 - 392:

“In summary, we demonstrated we can bias human choices using fast ultrasonic neuromodulation. Transcranial ultrasound over the frontal eye fields, a model circuit for human decision-making, induced **robust** facilitatory effects promoting contralateral saccades.”

That said, we would like to emphasize that the primary ‘control’ condition for the effects are the lateralized FEFs. The primary hypothesis is that stimulation of the left FEF should have opposite effects to stimulation of the right FEF. Indeed, as stated in the manuscript, our primary pre-registered comparison is left versus right FEF. We have made this more explicit in the manuscript.

Methods, lines 674-677:

“To measure the effects of TUS on saccade behavior, we first present our primary pre-registered comparison of left versus right FEF, **for which we expected opposing (lateralized) effects on saccade choices**: the lateralized FEFs are each other’s most informative control given their contralateral contributions to saccade selection.”

Note that the comparison to auditory sham would have been most important should we have observed perturbatory effects of FEF stimulation (i.e. FEF stimulation on both sides leads to more ‘random’ responses, rather than directional effects).

Finally, regarding the relatively weaker right FEF–sham contrast, one plausible explanation is that FEF-related circuitry appears less strongly lateralized in humans than in nonhuman primates, which could explain the weaker effects (also relative to a previous study in non-human primates, Kubanek et al. 2020). This is discussed in the manuscript (**Discussion, lines 367-371**):

“Second, the FEF circuitry in animals appears more lateralized compared to humans^{58,59}, which may serve to explain the somewhat weaker effect of particularly right FEF stimulation relative to sham. A left-right asymmetry in lateralization strength is plausible given the well documented right-hemisphere dominance of human attention networks and hemispheric differences in fronto-parietal circuitry⁶⁰⁻⁶³.”

One minor thing: Figure S12 – Should spatial-peak pressure amplitudes be 0.9 MPa, not kPa, based on the written description?

We thank the reviewer for noticing this. It should indeed be MPa (not kPa). This has now been corrected.